# Improving Generalization of Neural Combinatorial Optimization for Vehicle Routing Problems via Test-Time Projection Learning

**Yuanyao Chen**[1,2*], **Rongsheng Chen**[1,2,3*], **Fu Luo**[1,2], **Zhenkun Wang**[1,2†]

[1] School of Automation and Intelligent Manufacturing,
Southern University of Science and Technology, Shenzhen, China,

[2] Guangdong Provincial Key Laboratory of Fully Actuated System Control Theory and Technology,
Southern University of Science and Technology, Shenzhen, China,

[3] Peng Cheng Laboratory, Shenzhen, China,

12433020@mail.sustech.edu.cn, chenrs1017@gmail.com,
luof2023@mail.sustech.edu.cn, wangzhenkun90@gmail.com

## Abstract

Neural Combinatorial Optimization (NCO) has emerged as a promising learning-based paradigm for addressing Vehicle Routing Problems (VRPs) by minimizing the need for extensive manual engineering. While existing NCO methods, trained on small-scale instances (e.g., 100 nodes), have demonstrated considerable success on problems of similar scale, their performance significantly degrades when applied to large-scale scenarios. This degradation arises from the distributional shift between training and testing data, rendering policies learned on small instances ineffective for larger problems. To overcome this limitation, we introduce a novel learning framework driven by Large Language Models (LLMs). This framework learns a projection between the training and testing distributions, which is then deployed to enhance the scalability of the NCO model. Notably, unlike prevailing techniques that necessitate joint training with the neural network, our approach operates exclusively during the inference phase, obviating the need for model retraining. Extensive experiments demonstrate that our method enables a backbone model (trained on 100-node instances) to achieve superior performance on large-scale Traveling Salesman Problems (TSPs) and Capacitated Vehicle Routing Problems (CVRPs) with up to 100K nodes from diverse distributions. The source code can be found in https://github.com/CIAM-Group/TTPL.

## 1 Introduction

The vehicle routing problem (VRP) is prevalent in critical domains such as transportation [1], logistics [2], and supply chain management [3]. However, due to the NP-hard nature, solving a VRP can be complex and time-consuming [4]. While traditional exact and heuristic methods can yield optimal or near-optimal solutions, their high computational cost and dependence on domain expertise consequently diminish their feasibility for real-world deployment.

In recent years, neural combinatorial optimization (NCO) has emerged as a promising approach for solving VRP [5]. It utilizes neural networks to learn problem-solving strategies directly from the VRP instances, minimizing the need for algorithms designed by experts. Once trained, these neural networks can efficiently generate solutions for new instances at a low computational cost. Therefore,

---

[*]Equal contributors

[†]Corresponding author

39th Conference on Neural Information Processing Systems (NeurIPS 2025).

NCO methods demonstrate potential advantages in tackling limitations of traditional methods and exhibit promising performance in solving small-scale VRPs [6–9].

However, due to the different distributions between small and large-scale instances, the effectiveness of existing approaches degrades substantially when it comes to large-scale problems (e.g., problems with more than 1K nodes), thereby severely limiting their practical capability. To address the scalability issue, some attempts have been devoted to training NCO models on larger VRP instances (i.e., instances with 500 nodes) [10, 11]. However, the existing supervised learning (SL) and reinforcement learning (RL) both show their shortcomings when training on large-scale VRP instances. SL lacks sufficient labels (e.g., high-quality solutions) while RL suffers from extremely sparse rewards.

Consequently, current methods focus on training a model on small-scale instances and generalizing to large-scale scenarios via decomposition or local policy [12–15]. The decomposition-based method first simplifies large-scale problems to a series of subproblems. Subsequently, a solver trained on the small-scale instances can be utilized to construct a partial solution of the subproblems [16–18]. However, the scope of the decomposition often requires manual tuning, and even decomposing the problem could change its property, and thus damage the optimality of the solving algorithm. Another effort is local policy-based methods [14, 15]. It first reduces the search space to a small candidate subgraph at each construction step, typically based on the Euclidean distance to the last visited node. The next node is decided by the original policy or a local policy.

However, the distribution of selected subgraphs often differs from training instances, which significantly impairs the model's scalability. Current methods employ a projection technique to effectively transform these varied input distributions into a uniform distribution encountered during training [15, 13, 19]. Nevertheless, existing strategies require integration with the model training to ensure effectiveness during testing. Meanwhile, these manually designed strategies heavily rely on specialized domain expertise, thereby limiting their adaptability.

To address this challenge, we propose a novel learning framework called Test-Time Projection Learning (TTPL), driven by the Large Language Model (LLM), to design an efficient projection strategy. In particular, we utilize the LLM to learn the correlation between the input subgraph and training instances, thereby developing projection strategies that enhance model generalization. Different from existing works, our approach can be directly applied in the inference phase, does not need to train a model from scratch. Moreover, we propose a Multi-View Decision Fusion (MVDF) module to improve model generalization. Specifically, we perform data augmentation on the subgraph to generate multiple views. These views are subsequently processed by the model, with each view yielding a distinct node selection probability. Finally, these probabilities are aggregated, and the node with the highest resulting confidence score is selected.

Comprehensive experiments are conducted on both synthetic and real-world benchmarks for the TSP and CVRP. The results demonstrate that our proposed distribution adapter enables the base model, pre-trained on small-scale instances (e.g., 100 nodes), to achieve superior performance on the majority of large-scale VRP test instances without requiring additional fine-tuning. Our ablation study validates the effectiveness of the designed projection strategy.

## 2  Related Work

Existing NCO methods can be categorized into constructive [20–25] and non-constructive [26–29]. This paper mainly focuses on the constructive method; a detailed review of the non-constructive method and other LLM-based methods can be found in the Appendix. A.

### 2.1  Direct Generalization

This kind of method often trains neural networks on small-scale instances and generalizes them to large-scale instances. It is mostly known as the construction-based methods that predict an approximate solution in an autoregressive manner. Early studies demonstrate that neural models trained with SL or RL can reach promising performance on small-scale instances [30–32]. Moreover, [6, 33] propose to use the Transformer architecture [34] to design powerful attention-based models to solve VRPs. Subsequently, various Transformer-based approaches have been developed with different strengths [35–41]. Meanwhile, many studies focus on improving model performance on large-scale

VRPs [42]. In which [42] and [18] train the model with SL on 100-node and exhibit generalization ability to 1K nodes. Specifically, BQ [42] modifies the Markov Decision Process (MDP) to efficiently leverage the symmetries of combinatorial optimization problems (COPs). LEHD [18] develops a light encoder and heavy decoder structure to reach the same result. Nevertheless, the training distribution of 100-node instances drastically differs from the large-scale instances; the trained model usually performs poorly when tested on this large-scale scenario. Consequently, several attempts have trained models on larger-scale instances with up to 500 nodes [43, 11, 38, 10] or even directly training on the large-scale [44] (up to 100K-nodes). However, these approaches introduce prohibitive computational costs due to the exponentially growing search space.

## 2.2  Decomposition-Based Generalization

Apart from directly generalizing the model trained on a small-scale dataset to the large-scale instances, another line of research focuses on decomposing the large node graph into several small-scale subproblems. This approach involves solving them separately with a particular solver and then combining their solutions to form the solution of the large-scale problem [45–47, 12, 13, 48]. In particular, the problem decomposition and subproblem solving are often involved in different models. The task of the subproblem solver can be constructing a complete solution of a small-scale VRP or partial solutions of large-scale instances [16–18]. However, such decomposition introduces two critical shortcomings. When solving other complicated VRPs (e.g., CVRP), such decomposition becomes intractable and cannot be achieved by an individual strategy or model. Furthermore, when the decomposed subproblem is not small enough, it may still require other generalization techniques to solve.

## 2.3  Local Policy-Based Generalization

Another approach to solving large-scale VRPs is to reduce the search space to the $K$-Nearest-Neighbors (KNN) from the last visited node. The final decision is guided by either the original neural model with auxiliary distance information [49] or a local policy [14]. In addition, [15, 50, 42] directly select the candidate node from the neighborhood using NCO models. Although local policy can efficiently reduce the search space for constructive NCO, the reduced subgraph often drastically differs from the training instances, especially in large-scale or non-uniform instances. Consequently, the NCO models fail to distinguish the promising node. To this end, [15, 13, 19] propose a projection strategy to transform the extracted subgraph to a uniform distribution. While these strategies can enhance generalization, their effectiveness typically requires integration into model training, thereby imposing additional computational overhead and limiting their adaptability.

# 3  Preliminaries

**VRP Definition**  The VRP is defined on a graph $G = (V, E)$, where $V = \{v_0, v_1, ..., v_n\}$ represents nodes (with $v_0$ as the depot in CVRP) and $E$ denotes edges. Each non-depot node $V \setminus \{v_0\}$ has coordinates $s_i \in \mathbb{R}^2$ and a demand $d_i$, while $v_0$ has no demand. For the TSP, the objective is to find a single tour that explicitly visits all nodes once, minimizing the total Euclidean distance. In CVRP, vehicles with fixed capacity must deliver goods from the depot to customers, forming multiple routes that start and end at $v_0$. Each customer is visited once, and the cumulative demand per route cannot exceed the vehicle capacity. Solutions for both problems aim to minimize the total length of the tour, with feasibility requiring adherence to node visitation and capacity constraints.

**Automatic Heuristic Design (AHD)**  AHD aims to automatically discover high-performing heuristics $h$ for a target task $T$, such as combinatorial optimization. Formally, given a task $T$ with input space $X_T$, AHD seeks to identify an optimal heuristic $h^*$ from a heuristic space $H$ by maximizing the expected performance over instances $x \in X_T$:

$$h^* = \arg\max_{h \in H} \mathbb{E}_{x \sim X_T} f_T(x, h), \tag{1}$$

where $f_T(x, h)$ quantifies the effectiveness of heuristic $h$ on instance $x$. The heuristic space $H$ encompasses all feasible strategies that adhere to the constraints of $T$. This approach aligns with hyper-heuristic methodologies [51], which automate heuristic selection or generation without domain-specific handcrafting.

**LLM-Based AHD** LLM-based AHD integrates LLMs into the evolutionary search for a high-performance heuristic [52–55]. This paradigm leverages LLMs to generate and refine heuristics through natural language reasoning and code synthesis. Within an evolutionary computation framework, LLMs simulate mutation and crossover operations by modifying or combining existing heuristics. For example, given a parent heuristic, LLMs can introduce new algorithmic components, adjust parameters, or merge features from multiple candidates, guided by linguistic prompts that encode task-specific objectives and constraints. This iterative process balances the exploration of high-quality candidates, enabling efficient traversal of the heuristic space $H$.

# 4 Distribution Projection in Large-Scale VRPs

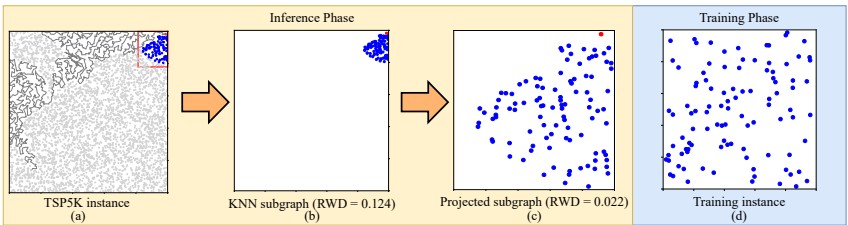

Figure 1: (a) Solution construction process for solving TSP5K instance, gray node, blue node, red node, and grey line denote the unvisited nodes, KNN ($k$=100) of the current node, current node, and constructed partial route, respectively. (b) Extracted KNN graph, RWD [56] indicates the distance between the input graph and a uniformly distributed training instance. (c) Projected KNN graph. (d) Training instance with 100 nodes.

When solving large-scale VRPs, current methods are facing challenges from the exponentially growing search space. As shown in the Figure. 1 (a), existing works reduce the search space to the KNN of the last visited node. Despite this reduction providing sufficient improvement, Figure. 1 (b) demonstrates that the selected KNN subgraphs come with various distributions that drastically differ from the training instance. Making it difficult for the model to predict the next promising node. Recent studies provide several projection techniques to project the input subgraph to a uniform distribution [15, 13, 19]. In this section, we systematically analyze the effectiveness of these projection methods and observe that such a technique has a great impact on the model's final performance.

Table 1: The effect of projection strategy on the gap in the LEHD model

| Method | TSP1K Gap | TSP5K Gap | TSP10K Gap | TSP50K Gap | TSP100K Gap |
|---|---|---|---|---|---|
| LEHD w/o proj | 4.45% | 12.80% | 17.42% | 48.00% | 121.42% |
| LEHD w/ proj | **2.96%** | **3.32%** | **3.77%** | **3.42%** | **3.24%** |

To validate the effectiveness of the projection, we employ the LEHD [18] as the base model, restricting its search space to the KNN ($k = 100$) nodes from the last visited node. Then, we apply the projection method used in INViT [15] to map the subgraph. Subsequently, the LEHD is adopted to select one promising node within the transformed subgraph. The experimental results are provided in Table 1. The experiment results exhibit that LEHD with the INViT projection method outperforms LEHD without projection consistently on all scales in TSP.

These findings underscore the criticality of maintaining distribution consistency between the model's training data and its inference inputs for effective large-scale generalization. Projection techniques serve as a primary mechanism to achieve this alignment by transforming varied input distributions to the training set. Consequently, the development of more sophisticated and robust projections is crucial for advancing model generalization capabilities.

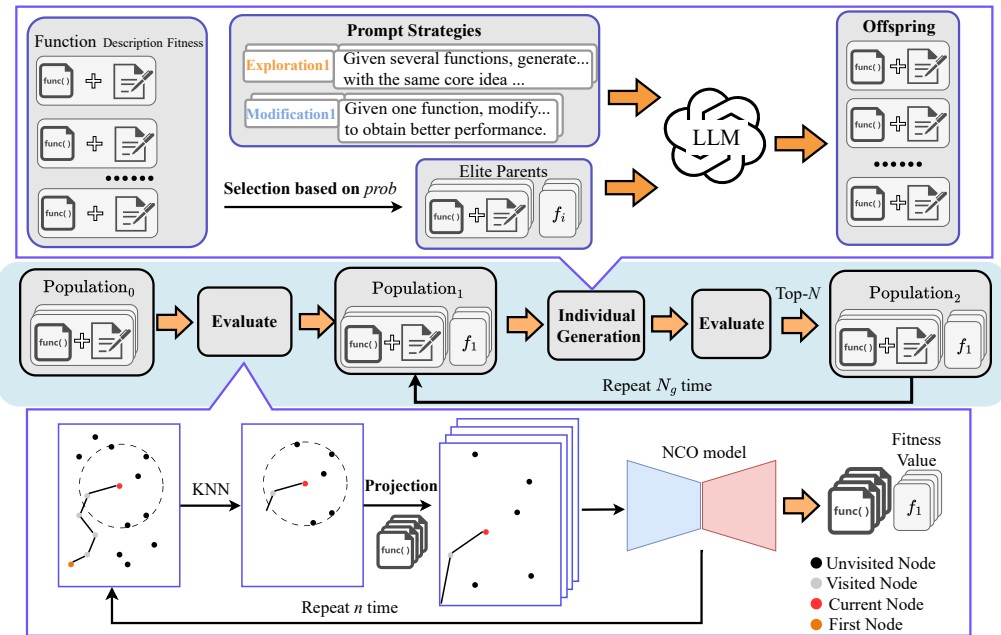

Figure 2: **The pipeline of the TTPL framework**. It comprises four components: initialization, fitness evaluation, offspring generation, and population update. **(a) Initialization:** TTPL establishes the initial population by generating individuals through prompting LLM with a predefined template. Following the initialization, an iterative optimization procedure is employed to search for the optimal individual. **(b) Offspring generation:** Offspring individuals are produced using several LLM-based evolutionary prompt strategies. **(c) Fitness evaluation:** An NCO model assesses the performance of these newly generated individuals. **(d) Population update:** The highest-performing individuals are then selected to constitute the succeeding generation, and this iterative process repeats until the specified termination criteria are satisfied.

## 5 Methodology

### 5.1 Test-Time Projection Learning

Current methods typically train models on small-scale, uniformly distributed datasets. However, these models often underperform on large-scale or real-world data due to the distributional shift between training and test instances. To address this challenge, we propose Test-Time Projection Learning (TTPL), an LLM-driven evolutionary framework that leverages LLM to generate an optimal projection strategy. Specifically, it comprises two components: 1) a NCO-based strategy evaluator is designed to measure the strategy generated by LLM; 2) an evolutionary projection strategy generator employs diverse prompt strategies to guide the LLM in developing these strategies.

**Method Framework**   The overall framework of TTPL is demonstrated in the Algorithm. 1. Its inputs consist of the population size $N$, number of generations $N_g$, the NCO model $f_\theta$, the LLM used to evolve the strategy, and the evaluation data set $D$. TTPL follows the general framework of EoH [52]. Therefore, we first generate $N$ individuals using LLM with the prompts from EoH. Here, individuals consist of a description in natural language and a code block that implements the projection. These individuals are then evaluated by the NCO model $f_\theta$ to construct the initial population $P_0$. Moreover, we conduct $N_g$ iterations to obtain an optimal projection strategy. In each iteration, we prompt LLMs to design different individuals based on a set of selected parents. Once the new population is generated, we select the first $N$ best individuals as the next generation. Finally, it outputs the optimal strategy it found in previous iterations

**Algorithm 1** $a^* = \textbf{TTPL}(N, N_g, f_\theta, \text{LLM}, D)$

---
1: **Input:**
2:     Population size: $N$; Number of Generation: $N_g$; NCO model: $f_\theta$;
3:     A given LLM; Evaluation dataset $D$
4: **Output**:
5:     Code for the best found projection strategy: $a^*$
6: **Initialization:**
7: **for** $j = 1 : N$ **do**
8:     Initialize new individual $a_j$ of the target model using LLM
9:     Evaluate $a_j$ and obtain the fitness value $f_\theta(a_j)$ with evaluation dataset $D$
10: **end for**
11:     Construct initial population $P_0 = \{a_1, \dots, a_N\}$
12: **for** $i = 1 : N_g$ **do**
13:     **for** $j = 1 : \text{N}$ **do**
14:         Generate new offspring using prompt strategies from [52]
15:         Evaluate offspring and obtain their fitness value with evaluation dataset $D$
16:     **end for**
17:     Construct offspring population $O$
18:     Sort $O$ in descending order based on their fitness value
19:     Update $P_{i-1}$ with the first $N$ individuals in $O$
20: **end for**
21: **Return:** The best found projection strategy: $a^*$;

---

**Individual Evaluation**    To evaluate the LLM-designed strategies, we deploy them within the target NCO model and calculate the objective value on the evaluation dataset $D$. Specifically, given a current node $v_i$, we first select $k$ nearest neighbors to form the subgraph $G_k$. Then, the strategy is performed on $G_k$ to obtain the projected graph $G_N$. Moreover, we input the projected subgraph to the given NCO model $f_\theta$ and select a node within $G_N$ as the destination of the next step. By repeating the above process, the final solution of a VRP instance is constructed step by step. Finally, the solution length is recorded as the fitness value of the input strategy.

**Individual Generation**    After initialization, $N_g$ iterations are conducted to evolve the population. In each iteration, we first randomly select a set of parent individuals from the current population based on the probability:

$$prob_i = \frac{1/2^{r_i}}{\sum_{j=1}^{N} 1/2^{r_j}}, \tag{2}$$

where $r_i$ is the rank of their fitness. After that, the selected parents are used to generate offspring through the four prompt strategies in [52], which consist of two groups, namely, Exploration (E1, E2) and Modification (M1, M2). The exploration prompt focuses on generating new individuals based on several selected parent strategies, while the modification prompt aims to modify the selected strategy for better performance. The details of four prompt strategies can be found in the Appendix. B.1

## 5.2   Multi-View Decision Fusion

Projection strategies transform the entire subgraph by a series of operations (e.g., rotation and translation) to align complex input distributions with the uniformly distributed training data. However, these graph-wise operations often struggle in normalizing local regions with density variations (i.e., strip-like node clusters in TSP), thereby impairing the model to distinguish promising nodes in local regions.

To resolve this limitation, we propose Multi-View Decision Fusion (MVDF), applied to the projected subgraphs. MVDF facilitates robust node identifications by generating multiple perspectives (views) of the subgraph. The model evaluates each perspective, and the resulting selection probabilities are aggregated. Ultimately, the perspective yielding the highest prediction score primarily guides the final decision.

In particular, given a subgraph with coordinates $\mathbf{X} \in \mathbb{R}^{k \times 2}$, we generate $M$ augmented variants through geometric transformations $\mathcal{T}_m(\mathbf{X})$ proposed in [6]. Each augmented subgraph $\mathbf{X}_m = \mathcal{T}_m(\mathbf{X})$

is solved by the NCO model to produce logits $\mathbf{l}_m \in \mathbb{R}^K$. The final selection probability is derived by aggregating logits across all augmentations:

$$\mathbf{p} = \sigma \left( \sum_{m=1}^{M} \mathbf{l}_m \right), \tag{3}$$

where $\sigma\left(\cdot\right)$ denotes the softmax function. This ensemble strategy forces the model to learn transformation-invariant features, effectively neutralizing local density deviations.

# 6 Experiment

Table 2: Comparison results on the synthetic TSP and CVRP dataset. *: Results are directly cited from the original publications. N/A: results that exceed the time limitation (seven days). OOM: results that exceed GPU memory limits.

| Method | TSP1K Obj.(Gap) | Time | TSP5K Obj.(Gap) | Time | TSP10K Obj.(Gap) | Time | TSP50K Obj.(Gap) | Time | TSP100K Obj.(Gap) | Time |
|---|---|---|---|---|---|---|---|---|---|---|
| LKH3 | 23.12 (0.00%) | 1.7m | 50.97 (0.00%) | 12m | 71.78 (0.00%) | 33m | 159.93 (0.00%) | 10h | 225.99 (0.00%) | 25h |
| Concorde | 23.12 (0.00%) | 1m | 50.95 (-0.05%) | 31m | 72.00 (0.15%) | 1.4h | N/A | N/A | N/A | N/A |
| Random Insertion | 26.11 (12.9%) | <1s | 58.06 (13.9%) | <1s | 81.82 (13.9%) | <1s | 182.65 (14.2%) | 15.4s | 258.13 (14.2%) | 1.7m |
| DIFUSCO* | 23.39 (1.17%) | 11.5s | —— | | 73.62 (2.58%) | 3.0m | —— | | —— | |
| H-TSP | 24.66 (6.66%) | 48s | 55.16 (8.21%) | 1.2m | 77.75 (8.38%) | 2.2m | OOM | | OOM | |
| GLOP | 23.78 (2.85%) | 10.2s | 53.15 (4.26%) | 1.0m | 75.04 (4.39%) | 1.9m | 168.09 (5.10%) | 1.5m | 237.61 (5.14%) | 3.9m |
| POMO aug * 8 | 32.51 (40.6%) | 4.1s | 87.72 (72.1%) | 8.6m | OOM | | OOM | | OOM | |
| ELG aug * 8 | 25.738 (11.33%) | 0.8s | 60.19 (18.08%) | 21s | OOM | | OOM | | OOM | |
| LEHD RRC1,000 | 23.29 (0.72%) | 3.3m | 54.43 (6.79%) | 8.6m | 80.90 (12.5%) | 18.6m | OOM | | OOM | |
| BQ bs16 | 23.43 (1.37%) | 13s | 58.27 (10.7%) | 24s | OOM | | OOM | | OOM | |
| SIGD bs16 | 23.36 (1.03%) | 17.3s | 55.77 (9.42%) | 30.5m | OOM | | OOM | | OOM | |
| INViT-3V greedy | 24.66 (6.90%) | 9.0s | 54.49 (6.90%) | 1.2m | 76.85 (7.07%) | 3.7m | 171.42 (7.18%) | 1.3h | 242.26 (7.20%) | 5.0h |
| LEHD greedy | 23.84 (3.11%) | 0.8s | 58.85 (15.46%) | 1.5m | 91.33 (27.24%) | 11.7m | OOM | | OOM | |
| BQ greedy | 23.65 (2.30%) | 0.9s | 58.27 (14.31%) | 22.5s | 89.73 (25.02%) | 1.0m | OOM | | OOM | |
| SIGD greedy | 23.573 (1.96%) | 1.2s | 57.19 (12.20%) | 15.5m | 93.80 (30.68%) | 15.5m | OOM | | OOM | |
| TTPL | 23.73 (2.65%) | 0.1s | 52.63 (3.25%) | 1.4s | 74.39 (3.63%) | 2.9s | 165.19 (3.29%) | 14.3s | 233.26 (3.22%) | 28.9s |
| TTPL-MVDF | 23.64 (2.26%) | 0.5s | 52.43 (2.86%) | 3.2s | 73.49 (2.39%) | 7.2s | 164.24 (2.69%) | 32.1s | 231.78 (2.56%) | 1.1m |
| TTPL-MVDF RRC1000 | **23.19 (0.28%)** | 3.6m | **51.50 (1.04%)** | 4.9m | **72.59 (1.13%)** | 5.0m | **162.40 (1.55%)** | 5.4m | **229.66 (1.63%)** | 5.9m |

| Method | CVRP1K Obj.(Gap) | Time | CVRP5K Obj.(Gap) | Time | CVRP10K Obj.(Gap) | Time | CVRP50K Obj.(Gap) | Time | CVRP100K Obj.(Gap) | Time |
|---|---|---|---|---|---|---|---|---|---|---|
| HGS | 41.2 (0.00%) | 5m | 126.2 (0.00%) | 5m | 227.3 (0.00) | 5m | 1081.0 (0.00%) | 4h | 2087.5 (0.00%) | 6.3h |
| GLOP-G (LKH-3) | 45.9 (11.4%) | 1.1s | 140.6(11.4%) | 4.0s | 256.4 (11.1%) | 6.2s | OOM | | OOM | |
| POMO aug * 8 | 101 (145.15%) | 4.6s | 632.9 (401.51%) | 11m | OOM | | OOM | | OOM | |
| ELG aug * 8 | 46.4 (12.62%) | 10.3s | OOM | | OOM | | OOM | | OOM | |
| LEHD RRC1,000 | **42.4 (2.91%)** | 3.4m | 132.7 (5.15%) | 10m | 243.8 (7.28%) | 51.6m | OOM | | OOM | |
| BQ bs16 | 43.1 (4.61%) | 14s | 136.4 (8.08%) | 2.4m | OOM | | OOM | | OOM | |
| SIGD bs16 | 44.3 (7.54%) | 22.5s | 137.5 (9.15%) | 2.7m | 247.2 (9.01%) | 6.0m | 1255.5 (16.30%) | 39.3m | OOM | |
| INViT-3V greedy | 48.2 (16.99%) | 11s | 146.6 (16.16%) | 1.4m | 262.1 (15.31%) | 4.3m | 1331.1 (23.1%) | 1.5h | 2683.4 (28.55%) | 5.8h |
| LEHD greedy | 44 (6.80%) | 0.8s | 138.2 (9.51%) | 1.4m | 256.3 (12.76%) | 12m | OOM | | OOM | |
| BQ greedy | 44.2 (7.28%) | 1s | 139.9 (10.86%) | 18.5s | 262.2 (15.35%) | 2m | OOM | | OOM | |
| SIGD greedy | 45.4 (10.39%) | 4.5s | 140.2 (11.31%) | 9.1s | 252.5 (11.34%) | 48.9s | 1274.3 (18.02%) | 10.2m | OOM | |
| TTPL | 44.7 (8.60%) | 0.1s | 134.4 (6.44%) | 0.5s | 238.9 (5.11%) | 3.8s | 1143.2 (5.72%) | 19.4s | 2220.4 (6.33%) | 41.3s |
| TTPL-MVDF | 44.4 (7.79%) | 0.5s | 132.4 (4.88%) | 2.5s | 233.0 (2.54%) | 7.7s | 1110.9 (2.73%) | 40.5s | 2163.5 (3.60%) | 1.3m |
| TTPL-MVDF RRC1000 | 42.5 (3.28%) | 4.1m | **129.2 (2.37%)** | 4.0m | **229.6 (1.02%)** | 6.4m | **1105.6 (2.25%)** | 7.0m | **2157.5 (3.32%)** | 7.7m |

**Problem Setting**    We evaluate our method on TSP and CVRP instances. Following established benchmarks [47, 18], the test sets include 128 instances for TSP1K and 16 instances each for TSP5K, 10K, 50K, and 100K. For CVRP, we use 100 instances for CVRP1K and CVRP5K and 16 instances each for CVRP10K, CVRP50K, and CVRP100K. CVRP capacities are 200 for CVRP1K and 300 for CVRP5K/10K/50K/100K. Optimal TSP solutions are calculated using LKH3 [57], while CVRP solutions are derived using HGS [58]. To further assess generalization, we follow [44] and evaluate on real-world instances from TSPLIB [59] and CVRPLIB [60], selecting symmetric EUC_2D instances with over 1K nodes (33 TSP and 14 CVRP instances). This ensures alignment with both synthetic benchmarks and practical scenarios.

**Model Setting**    Our experiments adopt LEHD as the base model [18], aligning with its original architecture and hyperparameters. The embedding dimension is configured to 128, and the decoder comprises 6 attention layers. Each multi-head attention layer uses 8 heads, while the feed-forward layer dimension is set to 512. We configure the KNN to 100. All experiments are performed on a single NVIDIA GeForce RTX 3090 GPU with 24GB of memory.

**LLM-AHD Setting**    The evolutionary framework follows the approach proposed by EoH [52]. The population size is set to 20, and the evolution process is run for 105 generations. The parent heuristic

is set to 2. The GPT-4o-mini model serves as the LLM backbone. For TSP/CVRP 1K, 5K, and 10K, we optimize one projection strategy for each scale, and generalize the strategy optimized on 10K to TSP/CVRP 50K and 100K due to the computational burden. At the beginning of the optimization, we input the projection strategy proposed in [15] as a seed to guide the heuristic design.

**Baselines** We compare TTPL with six categories of methods: **Classical Solvers:** Concorde [61], LKH3 [57] and HGS [58]; **Insertion Heuristic:** Random Insertion; **Construction-based NCO Methods:** POMO [7], BQ [42], LEHD [18], INVIT [15] and SIGD [62]; **Heatmap-based Methods:** DIFUSCO [28]; **Decomposition-based Method:** GLOP [13], H-TSP [12]; **Local Construction-based Method:** ELG [14].

**Metrics and Inference** We evaluate the performance using the optimality gap and inference time. The optimality gap quantifies the percentage difference between solutions generated by our method and the ground truth obtained from LKH3 (TSP) or HGS (CVRP). Inference time, reported in seconds (s), minutes (m), and hours (h), captures the computational efficiency of generating solutions across varying problem scales. For TTPL, we report three types of results, those obtained by greedy trajectory with a purely designed projection method, those obtained by greedy trajectory with our designed projection and MVDF, and those obtained with projection, MVDF, and Random Re-Construction (RRC) under 1000 iterations [18].

## 6.1 Experimental Results

**Results on Synthetic Dataset** Table 2 presents the comparative results, underscoring our method's superior performance and robust generalization capabilities across various problem scales. For TSP instances, our method, when relying on purely greedy construction, outperforms other greedy constructive methods on most problem scales, except TSP1K. However, by incorporating RRC for 1000 iterations, our approach consistently delivers the best performance across all TSP scales. Specifically, it reduces the optimality gap by 61.11%, 75.59%, 56.2%, 69.61%, and 68.29% for the TSP1K, 5K, 10K, 50K, and 100K instances, respectively, when compared to the leading alternative method. In the context of CVRP, our method, utilizing purely greedy sampling, secures the best results on all scales, with the exception of CVRP1K. Furthermore, by integrating RRC, our method is able to significantly surpass all other approaches across all evaluated CVRP scales. This translates to optimality gap reductions of 53.98%, 85.99%, 86.20%, and 88.37% on CVRP5K, 10K, 50K, and 100K instances, respectively, relative to the strongest competing baseline methods.

Table 3: Result on TSPLIB and CVRPLIB. OOM: results that exceed GPU memory limits. †: several instances are not solvable due to exceeding GPU memory limits.

| | TSPLIB | | | | CVRPLIB | | | |
|---|---|---|---|---|---|---|---|---|
| Method | 1K < n ≤ 5K | n > 5K | All | Solved # | 1K < n ≤ 7K | n > 7K | All | Solved # |
| GLOP | 5.02% | 6.87% † | 5.50% | 31/33 | 15.34% | 21.32% | 17.90% | 14/14 |
| ELG aug×8 | 11.34% | OOM | 11.34% | 23/33 | 10.57% | OOM | 10.57% | 6/14 |
| BQ bs16 | 10.65% | 30.58% † | 12.95% | 26/33 | 13.92% | OOM | 13.92% | 8/14 |
| LEHD RRC1,000 | 4.00% | 18.46% † | 7.37% | 30/33 | 8.42% | 21.53% † | 11.04% | 10/14 |
| SIGD greedy | 12.37% | 152.88% † | 48.63% | 31/33 | 14.73% | 49.49% | 29.63% | 14/14 |
| BQ greedy | 11.64% | 162.12% † | 64.65% | 32/33 | 16.92% | 52.27% | 32.07% | 14/14 |
| INViT greedy | 11.49% | 10.00% | 11.04% | 33/33 | 15.87% | 26.64% | 20.49% | 14/14 |
| LEHD greedy | 11.14% | 39.34% † | 17.72% | 30/33 | 15.20% | 32.80% † | 18.72% | 10/14 |
| TTPL | 6.04% | 4.57% | 5.80% | 33/33 | 11.89% | 14.93% | 13.19% | 14/14 |
| TTPL-MVDF | 3.81% | 4.05% | 3.88% | 33/33 | 10.41% | 11.71% | 10.97% | 14/14 |
| TTPL-MVDF RRC1000 | **1.04%** | **2.12%** | **1.37%** | 33/33 | **5.20%** | **10.56%** | **7.50%** | 14/14 |

**Results on Benchmark** We further test our method on TSPLIB [59] and CVRPLIB [60]. The detailed results are demonstrated in the Table 3. When using greedy construction for inference, LEHD purely with our designed projection can achieve a better result than other construction-based methods on all groups of instances. Using the MVDF and RRC further strengthens our performance, which yields a significant gap between the second-best methods.

## 6.2 Ablation Study

Table 4: Comparison between different projection strategies on the synthetic TSP dataset.

| | TSP1K Obj.(Gap) | Time | TSP5K Obj.(Gap) | Time | TSP10K Obj.(Gap) | Time | TSP50K Obj.(Gap) | Time | TSP100K Obj.(Gap) | Time |
|---|---|---|---|---|---|---|---|---|---|---|
| w/o proj | 24.2 (4.49%) | 0.1s | 57.5 (12.80%) | 1.4s | 84.3 (17.42%) | 2.7s | 236.7 (48.00%) | 13.2s | 500.4 (121.42%) | 27.8s |
| seed proj | 23.8 (2.96%) | 0.1s | 52.7 (3.32%) | 1.5s | 74.5 (3.77%) | 2.9s | 165.4 (3.42%) | 14.4s | 233.3 (3.24%) | 29.6s |
| TTPL | **23.73 (2.65%)** | 0.1s | **52.63 (3.25%)** | 1.4s | **74.39 (3.63%)** | 2.9s | **165.19 (3.29%))** | 14.3s | **233.26 (3.22%)** | 28.9s |

**Effects of Designed Projection**    To validate the effectiveness of the LLM-designed projection strategy, we design two variants of our method: 1) w/o proj: LEHD only uses KNN to reduce the search space during inference. 2) seed proj: The projection strategy is replaced with the input seed strategy in [15]. As exhibited in the Table. 4, the projection strategy designed from the TTPL framework consistently outperforms the two variants on all scales.

Table 5: Comparison between LEHD with MVDF, POMO aug, and without MVDF on synthetic TSP dataset

| | TSP1K Obj.(Gap) | Time | TSP5K Obj.(Gap) | Time | TSP10K Obj.(Gap) | Time | TSP50K Obj.(Gap) | Time | TSP100K Obj.(Gap) | Time |
|---|---|---|---|---|---|---|---|---|---|---|
| w/o MVDF | 23.7 (2.65%) | 0.1s | 52.6 (3.25%) | 1.4s | 74.4 (3.63%) | 2.9s | 165.2 (3.29%) | 14.3s | 233.3 (3.22%) | 28.9s |
| POMO aug | **23.4 (0.89%)** | 0.5s | **52.1 (2.22%)** | 3.1s | 73.6 (2.56%) | 6.3s | 164.5 (2.86%) | 36.6s | 232.4 (2.85%) | 1.5m |
| TTPL-MVDF | 23.6 (2.26%) | 0.5s | 52.4 (2.86%) | 3.2s | **73.5 (2.39%)** | 7.2s | **164.2 (2.69%)** | 32.1s | **231.8 (2.56%)** | 1.1m |

**Effects of MVDF**    To assess the efficacy of our proposed MVDF, we compare our complete method against two configurations: 1) w/o MVDF: LEHD utilizes the designed projection but without MVDF during inference; 2) POMO aug: test instances are first augmented using the POMO augmentation [7], and then solved with LEHD equipped with the designed projection. The complete results are shown in Table 5. Our MVDF approach outperforms the w/o MVDF variant in 5 out of 5 cases and the POMO aug variant in 3 out of 5 cases. Notably, MVDF demonstrates particular effectiveness when solving large-scale instances.

## 6.3   Versatility study

Table 6: Results of TTPL-MVDF designed strategy performed on POMO on randomly generated test instances

| | TSP1K Obj. | Time | TSP5K Obj. | Time | TSP10K Obj. | Time | TSP50K Obj. | Time | TSP100K Obj. | Time |
|---|---|---|---|---|---|---|---|---|---|---|
| POMO knn | 33.4 | 0.2s | 96.1 | 3.8s | 153.4 | 7.5s | 392.4 | 38.5s | 597.8 | 1.3m |
| POMO seed | 30.0 | 0.2s | 73.8 | 4.0s | 103.0 | 7.9s | 235.8 | 39.9s | 331.3 | 1.3m |
| POMO Ours | **29.8** | 1.3s | **67.6** | 7.8s | **94.6** | 15.5s | **212.1** | 1.3m | **296.6** | 2.6m |

**Versatility on the Other Model**    The versatility of our proposed techniques is further investigated by replacing the base model from LEHD with POMO, a well-known model trained by RL. We compare POMO, which is equipped with our proposed projection and MVDF technique, against POMO using KNN and the seed projection method. As shown in the Table 6, our method on POMO surpasses the compared approaches, indicating promising adaptability to different base models.

**Versatility on Different Distributions**    The robustness of TTPL-MVDF is evaluated on cross-distribution instances (cluster, explosion, and implosion) from [15]. Compared against LEHD without projection and LEHD with seed projection, TTPL-MVDF demonstrates a significant improvement in the base model's robustness. These findings (see Appendix D.2 for full results) validate the effectiveness of our proposed method in handling distributional shifts.

## 7   Conclusion, Limitation, and Future Work

In this work, we have presented an LLM-driven projection learning framework to improve model generalization in large-scale vehicle routing problems (VRPs). The core idea is to utilize an LLM to model the relationship between training and test data distribution, thereby informing the generation of an optimized projection strategy. To further boost performance, we have developed Multi-View

Decision Fusion (MVDF), which presents the model with multiple views of the original problem instance, enabling it to select the most confident one to guide node selection at each construction step. Empirical evaluation on diverse synthetic and real-world TSP and CVRP benchmarks confirms that our approach markedly enhances the zero-shot generalization capability of a base model (trained solely on 100-node instances) to large-scale instances.

**Limitation and Future Work**  While TTPL-MVDF shows superior performance on large-scale instances, a limitation of TTPL-MVDF is the relatively slow convergence to optimal strategies within its LLM optimization phase. In the future, we will investigate more efficient LLM optimization operators for improved evolution speed.

## Acknowledgments and Disclosure of Funding

This work was supported in part by the National Natural Science Foundation of China (Grant 62476118), the Natural Science Foundation of Guangdong Province (Grant 2024A1515011759), the Guangdong Science and Technology Program (Grant 2024B1212010002), and the Center for Computational Science and Engineering at Southern University of Science and Technology.

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

# A  Related work

## A.1  Review for Non-Constructive Method

Despite direct learning to construct a VRP solution, other methods work closely with search and improvement methods [63–65]. Early studies [66] utilize a graph convolutional network to generate heatmaps indicating the probability of edges belonging to optimal solutions for TSP instances. Subsequently, an approximate solution can be obtained via beam search [66], Monte Carlo tree search [19], dynamic programming [67], and guided local search [68]. While initially developed for small-scale problems, recent studies extend heatmap-based methods to larger TSP instances [19, 69]. However, the adaptability of these methods can be limited by their reliance on carefully tuned search strategies. Furthermore, the heatmap representation primarily focuses on edge probabilities, which may restrict applicability to problems with complex constraints.

Another research direction involves learning-augmented algorithms, where neural networks enhance traditional heuristic solvers. This includes accelerating solvers by learning operation selection [63, 70, 71], guided improvement [72–74], and problem decomposition [45, 75]. Many improvement-based methods propose to iteratively refine feasible solutions [76, 77, 70]. Nonetheless, these methods typically depend on well-designed heuristic operators or advanced solvers [78, 57] for solving different problems.

## A.2  Review for LLM-AHD Method

Recent advancements demonstrate the significant potential of Large Language Models (LLMs) in automating the design of high-performing heuristics for complex computational problems. This methodology typically involves maintaining a population of elite heuristic functions, which are evaluated and ranked based on their fitness on a designated dataset. The LLM is then iteratively prompted with high-performing functions from this population to generate potentially superior heuristic candidates. EoH [52] and Funsearch [79] pioneer the application of LLM to design a population-based evolutionary computation procedure. Building on this, ReEvo [80] incorporates a reflection mechanism to augment the LLM's reasoning capabilities during heuristic generation. Addressing the challenge of premature convergence, HSEvo [81] introduces diversity metrics and a harmony search algorithm to enhance population diversity without degrading solution quality. In a different approach, MCTS-AHD [53] conceptualizes the task as a search problem, modeling the heuristic design process as a Monte Carlo tree search to systematically explore the solution space.

# B  Detailed Methodology

In TSP instances, the input distribution can be interpreted as the coordinate distribution, and transforming the coordinate to a uniform distribution can be seen as normalization. To ease the understanding, we replace the projection with normalization when prompting the LLM. While the input of CVRP instances has extra demand and capacity, changing these features can easily lead to unfeasible solutions. Consequently, we limit the modification to input within custom node coordinates, and the projection can be similarly interpreted as normalization.

## B.1  Prompts of TTPL

Our LLM-driven prompt framework for projection evolution consists of four core strategies (E1, E2, M1, M2) [52], each structured around three components: **task description**, **strategy-specific instructions**, **parent projection**, and **template program**. In the following, we detail their design principles:

- **E1 (Exploratory Generation):** This prompt instructs an LLM to generate entirely new projection strategies by contrasting two existing approaches, explicitly avoiding structural similarities. It emphasizes divergence, requiring the LLM to innovate beyond the provided examples.

**Prompt for E1**

I need help designing an innovative coordinate normalization strategy function implemented in PyTorch to normalize a set of nodes' coordinates, aiming to maximize the final negative gap. The input is a tensor with shape (batch, num_nodes, 2) and the 'all_coors' must be a tensor with the same shape as 'coor1'.

I have 2 existing algorithms with their codes as follows:
No. 1 algorithm and the corresponding code are:
{projection description}
{Code}
No. 2 algorithm and the corresponding code are:
{projection description}
{Code}

Please help me create a new algorithm that has a completely different form from the given ones.
1. First, describe your new algorithm and main steps in one sentence. The description must be within boxed {}.
2. Next, implement the following Python function:
{Template Projection}

- **E2 (Backbone-Driven Innovation):** Focused on identifying common design principles across parent strategies, this prompt guides an LLM to extract shared computational patterns (e.g., scaling mechanisms, coordinate transformations) and generate variants that preserve these core ideas while introducing novel components.

**Prompt for E2**

I need help designing an innovative coordinate normalization strategy function implemented in PyTorch to normalize a set of nodes' coordinates, aiming to maximize the final negative gap. The input is a tensor with shape (batch, num_nodes, 2) and the 'all_coors' must be a tensor with the same shape as 'coor1'.

I have 2 existing algorithms with their codes as follows:
No. 1 algorithm and the corresponding code are:
{Projection description}
{Code}
No. 2 algorithm and the corresponding code are:
{Projection description}
{Code}

Please help me create a new algorithm that has a completely different form from the given ones, but can be motivated by them.
1. Firstly, identify the common backbone idea in the provided algorithms.
2. Secondly, based on the backbone idea, describe your new algorithm in one sentence. The description must be within boxed {}.
3. Thirdly, implement the following Python function:
{Template Projection}

- **M1 (Structural Mutation):** Targets architectural modifications by presenting a single parent strategy and prompting an LLM to redesign its core operations (e.g., changing anchor point for reference).

### Prompt for M1

I need help designing an innovative coordinate normalization strategy function implemented in PyTorch to normalize a set of nodes' coordinates, aiming to maximize the final negative gap. The input is a tensor with shape (batch, num_nodes, 2) and the 'all_coors' must be a tensor with the same shape as 'coor1'.

I have one algorithm with its code as follows. Algorithm description:
{Projection description}
Code:
{Code}

Please assist me in creating a new algorithm that has a different form but can be a modified version of the algorithm provided.
1. First, describe your new algorithm and main steps in one sentence. The description must be within boxed {}.
2. Next, implement the following Python function:
{Template Projection}

---

- **M2 (Parametric Optimization):** Specializes in parameter tuning for critical projection variables (e.g., scaling factors, shift coefficients), directing an LLM to systematically adjust numerical settings.

### Prompt for M2

I need help designing an innovative coordinate normalization strategy function implemented in PyTorch to normalize a set of nodes' coordinates, aiming to maximize the final negative gap. The input is a tensor with shape (batch, num_nodes, 2) and the 'all_coors' must be a tensor with the same shape as 'coor1'.

I have one algorithm with its code as follows. Algorithm description:
{Projection description}
Code:
{Code}

Please identify the main algorithm parameters and assist me in creating a new algorithm that has different parameter settings for the score function provided.
1. First, describe your new algorithm and main steps in one sentence. The description must be within boxed {}.
2. Next, implement the following Python function:
{Template Projection}

---

The prompt engineering for normalization evolution adopts a standardized template structure across strategies. The template code for TSP and CVRP builds on the INViT framework [15].

### Task Description for TSP

I need help designing an innovative coordinate projection strategy function implemented in PyTorch to normalize a set of nodes' coordinates, aiming to maximize the final negative gap. The input is a tensor with shape (batch, num_nodes, 2) and the 'all_coors' must be a tensor with the same shape as 'coor1'.

## Template Program for TSP

```python
def normalize(coor1: torch.Tensor) -> torch.Tensor:
    """
    Args:
        coor1: coordinates of nodes, shape: (batch, 1+k+1, 2)

    Return:
        all_coors: a tensor containing normalized coordinates, same shape as
    coor1

    Note:
        first_node: coor1[:,[0],:], left_node: coor1[:,1:-1,:], last_node:
    [:,[-1],:]
        left_node is the topk close to the last_node
    """
    batch_size = coor1.shape[0]
    all_coors = coor1
    graph = all_coors[:, 1:, :]
    min_values = torch.reshape(torch.min(graph, 1).values, (batch_size, 1, 2)
    )
    all_coors = all_coors - min_values  # translate
    ratio_x = torch.reshape(
        torch.max(graph[:, :, 0], 1).values - torch.min(graph[:, :, 0], 1).
    values,
        (-1, 1),
    )
    ratio_y = torch.reshape(
        torch.max(graph[:, :, 1], 1).values - torch.min(graph[:, :, 1], 1).
    values,
        (-1, 1),
    )
    ratio = torch.max(torch.cat((ratio_x, ratio_y), 1), 1).values
    ratio[ratio == 0] = 1
    all_coors = all_coors / (torch.reshape(ratio, (batch_size, 1, 1)))
    all_coors[ratio == 0, :, :] = (
        all_coors[ratio == 0, :, :] + min_values[ratio == 0, :, :]
    )
    all_coors = torch.clip(all_coors, 0, 1)
    return all_coors
```

## Task Description for CVRP

I need to develop a coordinate normalization method for CVRP sequences that preserves critical geometric relationships between nodes while enabling effective neural network processing. The function should take depot/vehicle coordinates (coor1), customer node (coor2), and final stop coordinates (coor3) as PyTorch tensors, maintaining their original shapes. Key objectives are: 1) Establish spatial consistency across batches without distorting relative positions, 2) Use the initial node as an anchor point for stable reference, 3) Prevent information loss from hard clipping while controlling magnitude variance, and 4) Ensure scale-invariant features that help the downstream model generalize across problem sizes. The solution should particularly focus on maintaining directional relationships and proportional distances rather than absolute positional constraints.

**Template Program for CVRP**

```python
def normalize(
    coor1: torch.Tensor, coor2: torch.Tensor, coor3: torch.Tensor
) -> torch.Tensor:
    """
    Args:
        coor1: indicate the first node, shape: (batch, 1, 2)
        coor2: coordinates of the rest of nodes, shape: (batch, 100, 2)
        coor3: coordinate of the last node, shape: (batch, 1, 2)

    Return:
        coor1: normalized coordinate of the first node, shape: (batch, 1, 2)
        coor2: normalized coordinates of the rest of nodes, shape: (batch,
    left_num, 2)
        coor3: normalized coordinate of the last node, shape: (batch, 1, 2)
    """
    lengths = [coor1.shape[1], coor2.shape[1], coor3.shape[1]]
    all_coors = torch.cat((coor1, coor2, coor3), dim=1)
    last_neighbors_xy = all_coors[:, 1:, :]
    # shape: (batch, 1+neighbor_k, 2)
    xy_max = torch.max(last_neighbors_xy, dim=1, keepdim=True).values
    xy_min = torch.min(last_neighbors_xy, dim=1, keepdim=True).values
    # shape: (batch, 1, 2)
    ratio = torch.max((xy_max - xy_min), dim=-1, keepdim=True).values
    ratio[ratio == 0] = 1
    # shape: (batch, 1, 1)
    all_coors = torch.clip((all_coors - xy_min) / ratio.expand(-1, 1, 2), 0,
    1)
    coor1, coor2, coor3 = torch.split(all_coors, lengths, dim=1)
    return coor1, coor2, coor3
```

## B.2 EoH Algorithm

The EoH is a population-based framework for LLM-based AHD [52]. The algorithm optimizes heuristics by iteratively applying LLM-guided mutation, crossover, and selection operations, enabling the exploration of diverse heuristic designs.

EoH begins by initializing a population of $N$ heuristics using LLM, where each heuristic is evaluated on dataset $D$ to compute its fitness value $f$. In each generation, five evolution strategies (E1, E2, M1, M2, M3) are used to generate new heuristics. For each strategy, $N$ parent heuristics are sampled from the current population using rank-based probabilities. LLMs then generate new heuristics by mutating or combining features from the parents, guided by the evolution strategy. New candidates are evaluated in $D$, and feasible ones are added to a candidate pool after removing duplicates. Finally, the top $N$ heuristics of the expanded pool are selected to form the next generation. This process repeats for $N_g$ generations, eventually returning the heuristic with the highest fitness. The pseudo-code for the EoH is provided in the Algorithm 2.

**Algorithm 2** Evolution of heuristics for automatic heuristic design

1: **Input:**
2:     Population size $N$, Max generations $N_g$, Evaluation dataset $D$, Evaluation function $f$.
3: **Output:**
4:     The best found heuristic $h^*$.
5: **Step 0: Initialization**
6: **for** $j = 1 : N$ **do**
7:     Generate heuristic $h_j$ using initialization prompt.
8:     Evaluate $h_j$ on $D$ to obtain fitness $f(D, h_j)$.
9: **end for**
10: Build initial population $P_0 \leftarrow \{h_1, ..., h_N\}$.
11: Set $O \leftarrow P_0$.
12: **Step 1: Generation of Heuristics**
13: **for** $i = 1 : N_g$ **do**
14:     Rank $P_{i-1}$ by fitness $\rightarrow \{r_1, ..., r_N\}$.
15:     **for** $S \in \{E1, E2, M1, M2, M3\}$ **do**
16:         **for** $j = 1 : N$ **do**
17:             Sample parents from $P_{i-1}$ via rank-based sampling.
18:             Generate new heuristic $h_j$ based parents using evolution strategies $S$.
19:             Evaluate $h_j$ on the evaluation dataset $D$ and obtain the fitness value $f_\theta(D, h_j)$.
20:             **if** $h_j$ is feasible **then**
21:                 Add $h_j$ to candidate pool $O$
22:             **end if**
23:         **end for**
24:     **end for**
25:     Remove duplicates in $O$.
26:     **Step 2: Population Management**
27:     Sort $O$ in descending order based on their fitness value.
28:     Update $P_i$ with the first $N$ heuristics in $O$.
29: **end for**
30: **return** Heuristic $h^*$ with the highest fitness in the final population.

# C TTPL Designed Strategies

## C.1 TSP Instances

In this section, we present representative projection strategies optimized by TTPL for TSP and CVRP instances. For TSP, the input is the coordinates of the nodes in the KNN subgraph, and the output is the corresponding transformed coordinates. For CVRP, the inputs consist of the coordinates of the depot, the current node, and the coordinates of the nodes in the KNN subgraph, and the output is the transformed coordinates of these nodes.

For TSP1K instances, TTPL evolves the following projection:

Code 1: Projection designed by TTPL on TSP1K

```python
def normalize(coor1: torch.tensor) -> torch.tensor:

    batch_size = coor1.shape[0]
    all_coors = coor1
    graph = all_coors[:, 1:, :]

    # Translate to the maximum values
    max_values = torch.reshape(torch.max(graph, dim=1).values, (batch_size, 1, 2))
    all_coors = max_values - all_coors  # translate

    # Calculate ranges for normalization
    ratio_x = torch.reshape(torch.max(graph[:, :, 0], dim=1).values - torch.min(
    graph[:, :, 0], dim=1).values, (-1, 1))
    ratio_y = torch.reshape(torch.max(graph[:, :, 1], dim=1).values - torch.min(
    graph[:, :, 1], dim=1).values, (-1, 1))

    # Find the maximum scale factor
    ratio = torch.max(torch.cat((ratio_x, ratio_y), dim=1), dim=1).values
    ratio[ratio == 0] = 1  # Avoid division by zero

    # Normalize the coordinates
    all_coors = all_coors / (torch.reshape(ratio, (batch_size, 1, 1)))
    all_coors[ratio == 0, :, :] = all_coors[ratio == 0, :, :] + max_values[ratio ==
    0, :, :]

    # Clip to ensure values are within [0, 1]
    all_coors = torch.clip(all_coors, 0, 1)

    return all_coors
```

The corresponding mathematical formulation proceeds as follows. Given input coordinates $S = \{s_0, s_1, \ldots, s_N\}$ where $s_i \in \mathbb{R}^2$, with $s_i^x$ and $s_i^y$ denoting the component of $s_i$, the projection process proceeds as follows:

$$M = \left( \max_{1 \leq j \leq N} s_j^x, \ \max_{1 \leq j \leq N} s_j^y \right).$$

The next step is to establish the maximum coordinate values $M = (M_x, M_y)$ as the reference point for transformations, which can be formulated as:

$$\widetilde{S} = \{\widetilde{s}_i \mid \widetilde{s}_i = M - s_i, \ \forall i \in \{0, \ldots, N\}\}.$$

Then, mirrored coordinates are created by reflecting points about the reference point $M$, ensuring all coordinates lie in the positive quadrant

$$r_x = \max_{1 \leq j \leq N} s_j^x - \min_{1 \leq j \leq N} s_j^x,$$

$$r_y = \max_{1 \leq j \leq N} s_j^y - \min_{1 \leq j \leq N} s_j^y.$$

Moreover, the coordinate ranges are calculated along both axes to determine the scaling factor

$$r = \begin{cases} \max(r_x, r_y) & \text{if } \max(r_x, r_y) > 0, \\ 1 & \text{otherwise.} \end{cases}$$

Such that, the maximum range is selected as the normalization factor, with special handling for zero-range edge cases

$$\widehat{S} = \left\{ \widehat{s}_i \,\middle|\, \widehat{s}_i = \frac{\widetilde{s}_i}{r} + M, \ \forall i \in \{0, \ldots, N\} \right\}.$$

Finally, coordinates are normalized to [0,1] relative to the reference point while preserving spatial relationships, and final coordinates are ensured to reside within the unit square through clipping operations, completing the projection

$$\widehat{S}_{clipped} = \left\{ \left(\min(\max(\widehat{s}_i^x, 0), 1), \min(\max(\widehat{s}_i^y, 0), 1)\right), \ \forall i \in \{0, \ldots, N\} \right\}.$$

For TSP5K instances, the evolved projection strategy is implemented as:

Code 2: Projection designed by TTPL on TSP5K

```python
def normalize(coor1: torch.tensor) -> torch.tensor:

    batch_size = coor1.shape[0]
    all_coors = coor1
    graph = all_coors[:, 1:, :]

    # Translate by the minimum values in the graph
    min_values = torch.reshape(torch.min(graph, 1).values, (batch_size, 1, 2))
    all_coors = all_coors - min_values  # translate

    # Apply a non-linear transformation
    all_coors = torch.tanh(all_coors)

    # Calculate scaling ratios with a slight modification
    ratio_x = torch.reshape(torch.max(graph[:, :, 0], 1).values - torch.min(graph[:,
      :, 0], 1).values, (-1, 1))
    ratio_y = torch.reshape(torch.max(graph[:, :, 1], 1).values - torch.min(graph[:,
      :, 1], 1).values, (-1, 1))
    ratio = torch.max(torch.cat((ratio_x, ratio_y), 1), 1).values

    # Avoid division by zero
    ratio[ratio == 0] = 1
    all_coors = all_coors / (torch.reshape(ratio, (batch_size, 1, 1)))

    # Post-process coordinates to ensure they are clipped within [0, 1]
    all_coors = torch.clip(all_coors, 0, 1)

    return all_coors
```

The projection process follows these operations:

$$M = \left( \min_{1 \leq j \leq N} s_j^x, \ \min_{1 \leq j \leq N} s_j^y \right)$$
$$\widetilde{S} = \{ \widetilde{s}_i \mid \widetilde{s}_i = s_i - M, \ \forall i \in \{0, \ldots, N\} \}$$
$$\widetilde{S} = \{ (\tanh(\widetilde{s}_i^x), \tanh(\widetilde{s}_i^y)), \ \forall i \in \{0, \ldots, N\} \}$$

The hyperbolic tangent function is employed to compress coordinates into (-1,1) while preserving spatial relationships. The detailed operations are listed as follows:

$$r_x = \max_{1 \leq j \leq N} s_j^x - \min_{1 \leq j \leq N} s_j^x,$$

$$r_y = \max_{1 \leq j \leq N} s_j^y - \min_{1 \leq j \leq N} s_j^y,$$

$$r = \begin{cases} \max(r_x, r_y) & \text{if } \max(r_x, r_y) > 0, \\ 1 & \text{otherwise,} \end{cases}$$

$$\widehat{S} = \left\{ \widehat{s}_i \ \middle| \ \widehat{s}_i = \frac{\widetilde{s}_i}{r}, \ \forall i \in \{0, \ldots, N\} \right\},$$

$$\widehat{S} = \left\{ (\min(\max(\widehat{s}_i^x, 0), 1), \min(\max(\widehat{s}_i^y, 0), 1)), \ \forall i \in \{0, \ldots, N\} \right\}.$$

For TSP10K instances, TTPL generates this projection:

Code 3: Projection designed by TTPL on TSP10K

```python
def normalize(coor1):

    batch_size = coor1.shape[0]
    all_coors = coor1.clone()
    graph = all_coors[:, 1:, :]

    # Step 1: Calculate midpoints for translation
    midpoints = (torch.max(graph, 1).values + torch.min(graph, 1).values) / 2
    midpoints = torch.reshape(midpoints, (batch_size, 1, 2))
    all_coors = all_coors - midpoints  # translate

    # Step 2: Calculate the new range after translation
    range_x = torch.reshape(torch.max(graph[:, :, 0], 1).values - torch.min(graph[:,
     :, 0], 1).values, (-1, 1))
    range_y = torch.reshape(torch.max(graph[:, :, 1], 1).values - torch.min(graph[:,
     :, 1], 1).values, (-1, 1))
    range_val = torch.max(torch.cat((range_x, range_y), 1), 1).values
    range_val[range_val == 0] = 1  # Prevent division by zero

    # Step 3: Scale coordinates
    all_coors = all_coors / (torch.reshape(range_val, (batch_size, 1, 1 )))

    # Step 4: Shift values to center in [0, 1]
    all_coors = (all_coors + 0.5)

    # Step 5: Clip the values to [0, 1]
    all_coors = torch.clamp(all_coors, 0, 1)

    return all_coors
```

It first calculates the centroid of the input coordinates using the average of extreme values:

$$M = \left( \frac{\max\limits_{1 \leq j \leq N} s_j^x + \min\limits_{1 \leq j \leq N} s_j^x}{2}, \frac{\max\limits_{1 \leq j \leq N} s_j^y + \min\limits_{1 \leq j \leq N} s_j^y}{2} \right).$$

Then, the projection is mathematically written below:

$$\widetilde{S} = \{\widetilde{s}_i \mid \widetilde{s}_i = s_i - M, \ \forall i \in \{0, \ldots, N\}\},$$

$$r_x = \max_{1 \leq j \leq N} s_j^x - \min_{1 \leq j \leq N} s_j^x,$$

$$r_y = \max_{1 \leq j \leq N} s_j^y - \min_{1 \leq j \leq N} s_j^y,$$

$$r = \begin{cases} \max(r_x, r_y) & \text{if } \max(r_x, r_y) > 0, \\ 1 & \text{otherwise}, \end{cases}$$

$$\widehat{S} = \left\{ \widehat{s}_i \ \middle| \ \widehat{s}_i = \frac{\widetilde{s}_i}{r}, \ \forall i \in \{0, \ldots, N\} \right\},$$

$$\widehat{S} = \widehat{S} + (0.5, 0.5),$$

$$\widehat{S}_{clipped} = \{(\min(\max(\widehat{s}_i^x, 0), 1), \min(\max(\widehat{s}_i^y, 0), 1)), \ \forall i \in \{0, \ldots, N\}\}.$$

## C.2 CVRP Instances

For CVRP instances, TTPL develops specialized strategies to handle depot-customer relationships. The CVRP1K projection strategy is implemented via:

Code 4: Projection designed by TTPL on CVRP1K

```python
def normalize(coor1: torch.Tensor, coor2: torch.Tensor, coor3: torch.Tensor) ->
    torch.Tensor:

    lengths = [coor1.shape[1], coor2.shape[1], coor3.shape[1]]
    all_coors = torch.cat((coor1, coor2, coor3), dim=1)

    # Calculate the relative vectors
    relative_vectors = all_coors - coor1

    # Compute the magnitudes of the vectors
    magnitudes = torch.norm(relative_vectors, dim=-1, keepdim=True)

    # Avoid division by zero
    magnitudes[magnitudes == 0] = 1

    # Normalize the vectors to unit vectors
    unit_vectors = relative_vectors / magnitudes

    # Scale unit vectors by the normalized magnitudes
    normalized_vectors = unit_vectors * (magnitudes / magnitudes.max(dim=1, keepdim=
    True).values)

    # Combine normalized vectors with the anchor point
    normalized_coors = coor1 + normalized_vectors

    coor1, coor2, coor3 = torch.split(normalized_coors, lengths, dim=1)

    return coor1, coor2, coor3
```

The projection first integrates all coordinates into a unified set with depot $s_0$ as the reference origin

$$v_i = s_i - s_0, \quad \forall i \in \{0, \ldots, N\}.$$

Secondly, position vectors are computed relative to the depot, establishing translation-invariant representations

$$\|v_i\| = \begin{cases} \sqrt{v_i^{x2} + v_i^{y2}} & \text{if } \sqrt{v_i^{x2} + v_i^{y2}} > 0, \\ 1 & \text{otherwise}. \end{cases}$$

Thirdly, vector magnitudes are computed with zero-division protection to ensure numerical stability

$$\widehat{v}_i = \frac{v_i}{\|v_i\|} \odot \frac{\|v_i\|}{\max\limits_{0 \leq j \leq N} \|v_j\|}.$$

Furthermore, magnitude scaling is performed relative to the maximum vector length

$$\widehat{s}_i = s_0 + \widehat{v}_i.$$

Then, absolute coordinates are reconstructed while maintaining spatial relationships

$$\widehat{S}_f = \{\widehat{s}_0\}, \quad \widehat{S}_k = \{\widehat{s}_1, \ldots, \widehat{s}_{N-1}\}, \quad \widehat{S}_l = \{\widehat{s}_N\}.$$

Finally, given coordinate sets $S_f = \{s_0\}$ (depot), $S_k = \{s_1, \ldots, s_{N-1}\}$, and $S_l = \{s_N\}$ where $s_i \in \mathbb{R}^2$, the normalized coordinates $\widehat{S}_f, \widehat{S}_k, \widehat{S}_l$ are computed through:

$$S = S_f \cup S_k \cup S_l = \{s_0, s_1, \ldots, s_N\}$$

For CVRP5K instances, the evolved projection strategy is implemented as:

Code 5: Projection designed by TTPL on CVRP5K

```python
def normalize(coor1: torch.Tensor, coor2: torch.Tensor, coor3: torch.Tensor) ->
    torch.Tensor:

    lengths = [coor1.shape[1], coor2.shape[1], coor3.shape[1]]
    all_coors = torch.cat((coor1, coor2, coor3), dim=1)

    # Get anchor point (coor1)
    anchor = coor1

    # Translate coordinates based on the anchor
    translated_coors = all_coors - anchor

    # Calculate distances to the anchor point
    distances = torch.norm(translated_coors, p=2, dim=-1, keepdim=True)

    # Find the furthest distance for scaling
    farthest_distance, _ = distances.max(dim=1, keepdim=True)
    scaling_factor = torch.sqrt(farthest_distance)

    scaling_factor[scaling_factor == 0] = 1  # Prevent division by zero

    # Normalize the coordinates using the scaling factor
    normalized_coors = translated_coors / scaling_factor.expand_as(translated_coors)

    # Combine back with the anchor
    normalized_coors += anchor

    # Split back to original coordinates
    coor1, coor2, coor3 = torch.split(normalized_coors, lengths, dim=1)

    return coor1, coor2, coor3
```

The projection steps are formalized by:

$$v_i = s_i - s_0, \quad \forall i \in \{0, \ldots, N\},$$

$$\|v_i\| = \sqrt{v_i^{x2} + v_i^{y2}},$$

$$v_{\max} = \begin{cases} \sqrt{\max_{0 \leq j \leq N} \|v_j\|} & \text{if } \sqrt{\max_{0 \leq j \leq N} \|v_j\|} > 0, \\ 1 & \text{otherwise,} \end{cases}$$

$$\widehat{v}_i = \frac{v_i}{v_{\max}},$$

$$\widehat{s}_i = s_0 + \widehat{v}_i,$$

$$\widehat{S}_f = \{\widehat{s}_0\}, \quad \widehat{S}_k = \{\widehat{s}_1, \ldots, \widehat{s}_{N-1}\}, \quad \widehat{S}_l = \{\widehat{s}_N\},$$

$$S = S_f \cup S_k \cup S_l = \{s_0, s_1, \ldots, s_N\}.$$

For CVRP10K instances, TTPL generates this projection:

Code 6: Projection designed by TTPL on CVRP10K

```python
def normalize(coor1: torch.Tensor, coor2: torch.Tensor, coor3: torch.Tensor) ->
    torch.Tensor:

    lengths = [coor1.shape[1], coor2.shape[1], coor3.shape[1]]
    all_coors = torch.cat((coor1, coor2, coor3), dim=1)

    # Centering the coordinates around the first node
    center = coor1.squeeze(1)  # shape: (batch, 2)
    relative_coors = all_coors - center.unsqueeze(1)  # shape: (batch, 1 + k, 2)

    # Calculate distances from the first node
    distances = torch.norm(relative_coors, dim=-1, keepdim=True)  # shape: (batch, 1
     + k, 1)

    # Apply a non-linear transformation to the distances (e.g., exponential scaling)
    transformed_distances = torch.exp(distances) - 1  # shape: (batch, 1 + k, 1)

    # Scale transformed distances to [0, 1]
    max_distance = torch.max(transformed_distances, dim=1, keepdim=True).values
    max_distance[max_distance == 0] = 1  # Prevent division by zero
    normalized_distances = transformed_distances / max_distance  # shape: (batch, 1
    + k, 1)

    # Maintain direction by re-constructing relative coordinates
    direction = relative_coors / (distances + 1e-6)  # shape: (batch, 1 + k, 2)
    normalized_coors = normalized_distances * direction  # shape: (batch, 1 + k, 2)

    all_coors = normalized_coors + center.unsqueeze(1)  # Translate back to original
     position

    coor1, coor2, coor3 = torch.split(all_coors, lengths, dim=1)

    return coor1, coor2, coor3
```

The projection process follows these operations:

$$v_i = s_i - s_0, \quad \forall i \in \{0, \ldots, N\},$$

$$\|v_i\| = \sqrt{v_i^{x\,2} + v_i^{y\,2}},$$

$$\widetilde{v}_i = e^{\|v_i\|} - 1,$$

$$v_{\max} = \begin{cases} \max\limits_{0 \le j \le N} \widetilde{v}_j & \text{if } \max\limits_{0 \le j \le N} \widetilde{v}_j > 0, \\ 1 & \text{otherwise,} \end{cases}$$

$$v_i = \frac{\widetilde{v}_i}{v_{\max}},$$

$$\widehat{v}_i = \frac{\widetilde{v}_i}{v_{\max}} \odot \frac{v_i}{\|v_i\| + 1e6},$$

$$\widehat{s}_i = s_0 + \widehat{v}_i,$$

$$\widehat{S}_f = \{\widehat{s}_0\}, \quad \widehat{S}_k = \{\widehat{s}_1, \ldots, \widehat{s}_{N-1}\}, \quad \widehat{S}_l = \{\widehat{s}_N\},$$

$$S = S_f \cup S_k \cup S_l = \{s_0, s_1, \ldots, s_N\}.$$

# D Experiment Details

## D.1 TSPLIB and CVRPLIB Results

We evaluate the proposed method in four benchmark scenarios: TSPLIB [59] and CVRPLib [60] instances with comparisons in both greedy and post-search frameworks. We compare our method against five baselines: INViT [15], SIGD [62], BQ [42], LEHD [18] and GLOP [13]. For TSPLIB, all experiments are conducted on large-scale instances ranging from 1,000 to 85,900 nodes to assess scalability and generalization.

As shown in the Table. 7, our method achieves a 3.88% average optimality gap across all 33 instances, significantly reducing the optimality gap by 64.86%, 92.02%, 94.00%, and 78.10% compared with INViT, SIGD greedy, BQ greedy, and LEHD greedy.

Table 7: Optimality gaps of greedy-based NCO methods on TSPLib instances. "OOM" signifies out-of-memory failures. Solved# and average gaps are summarized at the bottom.

| Instance | Scale | INViT | SIGD greedy | BQ greedy | LEHD greedy | Ours |
|---|---|---|---|---|---|---|
| dsj1000 | 1,000 | 9.20% | 7.82% | 6.96% | 8.31% | **2.40%** |
| pr1002 | 1,002 | 16.43% | 2.95% | 3.35% | 4.44% | **0.90%** |
| u1060 | 1,060 | 12.29% | 7.60% | 10.72% | 10.00% | **1.22%** |
| vm1084 | 1,084 | 10.91% | **3.38%** | 4.41% | 5.42% | 4.32% |
| pcb1173 | 1,173 | 8.67% | 4.32% | 5.61% | 8.01% | **2.25%** |
| d1291 | 1,291 | 22.37% | 4.50% | 10.93% | 14.13% | **3.59%** |
| rl1304 | 1,304 | 8.90% | **3.18%** | 8.48% | 8.14% | 3.76% |
| rl1323 | 1,323 | 17.02% | **3.64%** | 5.44% | 9.26% | 5.56% |
| nrw1379 | 1,379 | 5.25% | 27.91% | 18.96% | 15.49% | **1.00%** |
| fl1400 | 1,400 | 18.30% | 24.87% | 16.73% | 18.80% | **7.91%** |
| u1432 | 1,432 | 6.16% | **2.79%** | 4.65% | 7.96% | 2.81% |
| fl1577 | 1,577 | 12.63% | 22.37% | 19.95% | 14.68% | **4.97%** |
| d1655 | 1,655 | 12.81% | 14.55% | 12.53% | 13.89% | **5.57%** |
| vm1748 | 1,748 | 16.27% | 9.95% | 7.71% | 10.10% | **2.37%** |
| u1817 | 1,817 | 7.31% | 7.29% | 8.40% | 10.32% | **4.72%** |
| rl1889 | 1,889 | 10.44% | 7.30% | 7.93% | 7.49% | **5.70%** |
| d2103 | 2,103 | 16.79% | 10.46% | 16.48% | 14.57% | **5.78%** |
| u2152 | 2,152 | 10.38% | 10.22% | 11.56% | 12.65% | **4.01%** |
| u2319 | 2,319 | 0.98% | 3.73% | 4.33% | 4.18% | **0.31%** |
| pr2392 | 2,392 | 8.29% | 8.88% | 13.96% | 12.33% | **3.53%** |
| pcb3038 | 3,038 | 6.85% | 11.01% | 17.33% | 13.44% | **3.55%** |
| fl3795 | 3,795 | 18.75% | 57.46% | 30.97% | 13.55% | **9.29%** |
| fnl4461 | 4,461 | 7.32% | 28.33% | 20.35% | 19.05% | **2.16%** |
| rl5915 | 5,915 | 14.02% | 44.31% | 26.77% | 24.17% | **4.48%** |
| rl5934 | 5,934 | 12.91% | 53.73% | 33.19% | 24.11% | **4.26%** |
| pla7397 | 7,397 | 9.45% | 90.16% | 69.34% | 40.94% | **5.42%** |
| rl11849 | 11,849 | 12.71% | 105.26% | 46.65% | 38.04% | **4.86%** |
| usa13509 | 13,509 | 13.44% | 525.78% | 676.67% | 71.10% | **4.55%** |
| brd14051 | 14,051 | 9.31% | 138.65% | 145.41% | 41.22% | **3.98%** |
| d15112 | 15,112 | 7.24% | 145.56% | 172.38% | 35.82% | **2.09%** |
| d18512 | 18,512 | 6.62% | 119.58% | 126.51% | OOM | **1.97%** |
| pla33810 | 33,810 | 7.04% | OOM | 504.13% | OOM | **5.34%** |
| pla85900 | 85,900 | 7.21% | OOM | OOM | OOM | **3.54%** |
| Sloved# | | 33/33 | 31/33 | 32/33 | 30/33 | 33/33 |
| Avg.gap | | 11.04% | 48.63% | 64.65% | 17.72% | **3.88%** |

As shown in the Table 8, our method achieves a 1.37% average optimality gap across all 33 TSPLIB instances under the post-search framework, reducing the gap by 88.03%, 89.43%, 81.41%, and 75.09% compared to SIGD bs16, BQ bs16, LEHD RRC1000, and GLOP, respectively.

Table 8: Optimality gaps of post-search-based NCO methods on TSPLIB instances.

| Instance | Scale | SIGD bs16 | BQ bs16 | LEHD RRC1000 | GLOP | Ours RRC1000 |
|---|---|---|---|---|---|---|
| dsj1000 | 1,000 | 9.27% | 4.35% | 2.45% | 2.45% | **0.63%** |
| pr1002 | 1,002 | 2.25% | 2.04% | 1.08% | 3.02% | **0.14%** |
| u1060 | 1,060 | 5.45% | 5.82% | 2.79% | 2.40% | **0.27%** |
| vm1084 | 1,084 | 6.68% | 5.48% | 1.47% | 3.83% | **0.24%** |
| pcb1173 | 1,173 | 2.61% | 4.45% | 2.99% | 5.92% | **0.63%** |
| d1291 | 1,291 | 6.85% | 7.38% | 2.88% | 5.78% | **1.16%** |
| rl1304 | 1,304 | 3.64% | 6.30% | 3.14% | 8.02% | **0.12%** |
| rl1323 | 1,323 | 2.24% | 4.51% | 1.16% | 6.27% | **0.51%** |
| nrw1379 | 1,379 | 17.82% | 13.23% | 8.34% | 3.34% | **0.39%** |
| fl1400 | 1,400 | 33.75% | 37.48% | 2.96% | 2.15% | **1.14%** |
| u1432 | 1,432 | 2.10% | 4.25% | 2.24% | 3.40% | **0.67%** |
| fl1577 | 1,577 | 33.27% | 18.32% | 3.81% | 5.85% | **1.19%** |
| d1655 | 1,655 | 12.98% | 10.20% | 5.09% | 5.33% | **2.36%** |
| vm1748 | 1,748 | 4.59% | 6.42% | 3.36% | 3.75% | **0.39%** |
| u1817 | 1,817 | 5.91% | 6.07% | 4.83% | 6.26% | **2.61%** |
| rl1889 | 1,889 | 7.39% | 7.24% | 3.17% | 7.06% | **1.25%** |
| d2103 | 2,103 | 12.66% | 15.46% | 2.14% | 9.58% | **0.13%** |
| u2152 | 2,152 | 7.11% | 7.69% | 6.22% | 7.32% | **1.77%** |
| u2319 | 2,319 | 1.42% | 3.17% | 1.47% | 1.29% | **0.19%** |
| pr2392 | 2,392 | 5.68% | 10.53% | 5.63% | 5.20% | **0.76%** |
| pcb3038 | 3,038 | 6.56% | 11.51% | 7.49% | 5.28% | **1.32%** |
| fl3795 | 3,795 | 53.43% | 38.33% | 6.85% | 7.45% | **5.05%** |
| fnl4461 | 4,461 | 19.22% | 14.68% | 10.37% | 4.47% | **1.10%** |
| rl5915 | 5,915 | OOM | 19.58% | 12.38% | 10.35% | **1.76%** |
| rl5934 | 5,934 | OOM | 24.53% | 11.94% | 10.28% | **1.99%** |
| pla7397 | 7,397 | OOM | 47.63% | 15.31% | 5.94% | **2.68%** |
| rl11849 | 11,849 | OOM | OOM | 18.19% | 8.69% | **2.48%** |
| usa13509 | 13,509 | OOM | OOM | 31.37% | 5.32% | **1.85%** |
| brd14051 | 14,051 | OOM | OOM | 21.19% | 4.73% | **2.47%** |
| d15112 | 15,112 | OOM | OOM | 18.83% | 4.67% | **1.06%** |
| d18512 | 18,512 | OOM | OOM | OOM | 4.98% | **1.15%** |
| pla33810 | 33,810 | OOM | OOM | OOM | OOM | **3.37%** |
| pla85900 | 85,900 | OOM | OOM | OOM | OOM | **2.43%** |
| Sloved# | | 23/33 | 26/33 | 30/33 | 31/33 | 33/33 |
| Avg.gap | | 11.43% | 12.95% | 7.37% | 5.50% | **1.37%** |

For CVRPLIB, we evaluate both greedy and post-search frameworks on large-scale instances ranging from 1,000 to 30,000 nodes. As shown in the Table. 9 and Table. 10, our method achieves 10.97% and 7.34% average optimality gaps in the greedy and post-search frameworks, respectively, outperforming all baselines in both settings.

Table 9: Optimality gaps of greedy-based NCO methods on CVRPLIB instances.

| Instance | Scale | INViT | SIGD greedy | BQ greedy | LEHD greedy | Ours |
|---|---|---|---|---|---|---|
| X-n1001-k43 | 1,000 | 12.60% | 8.02% | 4.99% | 7.57% | **4.95%** |
| Li_30 | 1,040 | **9.12%** | 9.81% | 10.62% | 12.54% | 12.84% |
| Li_31 | 1,120 | 12.42% | 12.88% | 12.93% | **4.95%** | 16.12% |
| Li_32 | 1,200 | 9.90% | 13.61% | 13.48% | **7.68%** | 11.30% |
| Leuven1 | 3,000 | 13.71% | 16.19% | 18.53% | 16.60% | **5.69%** |
| Leuven2 | 4,000 | 26.08% | 25.64% | 30.70% | 34.85% | **14.99%** |
| Antwerp1 | 6,000 | 15.40% | 13.98% | 16.48% | 14.66% | **5.76%** |
| Antwerp2 | 7,000 | 27.75% | 17.72% | 27.67% | 22.77% | **11.66%** |
| Ghent1 | 10,000 | 15.87% | 24.22% | 25.36% | 27.23% | **5.73%** |
| Ghent2 | 11,000 | 30.78% | 28.09% | 54.04% | 38.36% | **15.56%** |
| Brussels1 | 15,000 | 18.09% | 25.48% | 35.70% | OOM | **6.80%** |
| Brussels2 | 16,000 | 32.08% | 38.32% | 50.90% | OOM | **15.38%** |
| Flanders1 | 20,000 | 23.41% | 43.95% | 37.51% | OOM | **7.35%** |
| Flanders2 | 30,000 | 39.60% | 136.89% | 110.09% | OOM | **19.43%** |
| Sloved# | | 14/14 | 14/14 | 14/14 | 10/14 | 14/14 |
| Avg.gap | | 20.49% | 29.63% | 32.07% | 18.72% | **10.97%** |

Table 10: Optimality gaps of post-search-based NCO methods on CVRPLIB instances.

| Instance | Scale | SIGD bs16 | BQ bs16 | LEHD RRC1000 | GLOP | Ours RRC1000 |
|---|---|---|---|---|---|---|
| X-n1001-k43 | 1,000 | 5.68% | 3.07% | **2.92%** | 16.78% | 3.52% |
| Li_30 | 1,040 | 9.04% | 11.86% | **3.28%** | 11.10% | 4.46% |
| Li_31 | 1,120 | 11.53% | 8.18% | 3.47% | 17.06% | **2.73%** |
| Li_32 | 1,200 | 9.40% | 7.43% | **1.45%** | 9.44% | 3.41% |
| Leuven1 | 3,000 | 14.80% | 15.39% | 10.71% | 14.95% | **3.54%** |
| Leuven2 | 4,000 | 22.82% | 25.69% | 21.22% | 16.54% | **10.61%** |
| Antwerp1 | 6,000 | 11.66% | 13.64% | 8.91% | 19.08% | **4.44%** |
| Antwerp2 | 7,000 | 16.27% | 26.09% | 15.42% | 17.72% | **8.91%** |
| Ghent1 | 10,000 | 21.36% | OOM | 17.28% | 18.28% | **4.89%** |
| Ghent2 | 11,000 | 27.71% | OOM | 25.77% | 16.45% | **13.39%** |
| Brussels1 | 15,000 | 23.12% | OOM | OOM | 26.17% | **6.26%** |
| Brussels2 | 16,000 | 34.87% | OOM | OOM | 17.56% | **13.71%** |
| Flanders1 | 20,000 | 40.92% | OOM | OOM | 24.02% | **6.90%** |
| Flanders2 | 30,000 | 129.81% | OOM | OOM | 25.42% | **18.18%** |
| Sloved# | | 14/14 | 8/14 | 10/14 | 14/14 | 14/14 |
| Avg.gap | | 27.07% | 13.92% | 11.04% | 17.90% | **7.50%** |

## D.2 Result in Cross-Distributions

To evaluate robustness against distribution shifts, we tested TTPL on TSP and CVRP instances under three distinct spatial distributions (Clustered, Explosion, and Implosion)[15] on scales from 1,000 to 100,000 nodes.

Table 11: Solution length and runtimes for TSP and CVRP instances under Clustered, Explosion, and Implosion distributions.

| | Clustered | | | | | | | | | |
|---|---|---|---|---|---|---|---|---|---|---|
| Method | TSP1K | | TSP5K | | TSP10K | | TSP50K | | TSP100K | |
| w/o proj | 15.3 | 0.1s | 35.4 | 1.4s | 57.1 | 2.7s | 185.6 | 13.5s | 277.4 | 27.6s |
| seed proj | 14.5 | 0.1s | 29.8 | 1.5s | 41.4 | 2.9s | 91.0 | 14.6s | 124.9 | 29.6s |
| TTPL | **14.4** | 0.5s | **29.6** | 3.2s | **41.3** | 6.4s | **90.5** | 33.9s | **123.9** | 1.1m |
| | Explosion | | | | | | | | | |
| | TSP1K | | TSP5K | | TSP10K | | TSP50K | | TSP100K | |
| w/o proj | 17.5 | 0.1s | 39.0 | 1.3s | 55.6 | 2.6s | 193.6 | 13.5s | 269.5 | 27.6s |
| seed proj | 16.7 | 0.1s | 34.1 | 1.5s | 43.1 | 2.9s | 88.4 | 14.6s | 111.8 | 29.5s |
| TTPL | **16.4** | 0.5s | **33.9** | 3.2s | **42.4** | 6.6s | **87.9** | 34.1s | **110.9** | 1.1m |
| | Implosion | | | | | | | | | |
| | TSP1K | | TSP5K | | TSP10K | | TSP50K | | TSP100K | |
| w/o proj | 21.1 | 0.1s | 52.5 | 1.3s | 75.9 | 2.6s | 212.0 | 13.5s | 428.1 | 27.5s |
| seed proj | 20.9 | 0.1s | **46.7** | 1.5s | 64.6 | 2.9s | 141.6 | 14.5s | 198.4 | 29.4s |
| TTPL | **20.7** | 0.5s | 46.7 | 3.2s | **64.2** | 6.6s | **140.6** | 33.8s | **196.8** | 1.1m |
| | Clustered | | | | | | | | | |
| Method | CVRP-1K | | CVRP-5K | | CVRP-10K | | CVRP-50K | | CVRP-100K | |
| w/o proj | 31.4 | 0.1s | 87.4 | 1.7s | 112.6 | 3.2s | 433.6 | 16.5s | 831.9 | 36.0s |
| seed proj | 37.4 | 0.1s | 90.3 | 1.7s | 109.1 | 3.4s | 217.7 | 17.2s | **373.7** | 40.7s |
| TTPL | **31.1** | 0.5s | **80.7** | 3.8s | **93.8** | 7.6s | **203.0** | 39.3s | 384.6 | 1.4m |
| | Explosion | | | | | | | | | |
| | CVRP-1K | | CVRP-5K | | CVRP-10K | | CVRP-50K | | CVRP-100K | |
| w/o proj | 30.0 | 0.1s | 70.2 | 1.6s | 90.3 | 3.2s | 401.3 | 16.5s | 744.9 | 35.9s |
| seed proj | 39.3 | 0.1s | 78.4 | 1.7s | 94.8 | 3.3s | 203.7 | 17.2s | 322.6 | 41.0s |
| TTPL | **29.6** | 0.5s | **64.3** | 3.8s | **75.5** | 7.7s | **178.6** | 39.1s | **294.4** | 1.4m |
| | Implosion | | | | | | | | | |
| | CVRP-1K | | CVRP-5K | | CVRP-10K | | CVRP-50K | | CVRP-100K | |
| w/o proj | 39.0 | 0.1s | 102.6 | 1.6s | 131.1 | 3.2s | 402.4 | 16.5s | 808.7 | 35.9 |
| seed proj | 51.5 | 0.1s | 124.3 | 1.7s | 149.6 | 3.3s | 325.2 | 17.2s | 549.7 | 41.7s |
| TTPL | **38.6** | 0.5s | **98.2** | 3.8s | **120.7** | 7.7s | **299.0** | 39.1s | **542.3** | 1.4m |

# E  Additional Experiment Analysis

## E.1  Adaptability of TTPL

To further validate the effectiveness of TTPL framework, we utilize our proposed framework on different base models. Specifically, we compare POMO [7], LEHD [18], BQ [42], and SIGD [62], four **constructive** models, with and without TTPL projection. The test instances are set to TSP1K/5K/10K with the same settings as the previous. The results are shown in the Table. 12

## E.2  TTPL Time Usage

To quantify the extra computational overhead introduced by the TTPL, we report the training time used for TTPL to search for a projection strategy in Table 13

## E.3  Results from Other LLMs

To investigate the robustness of TTPL to different LLMs, we test our framework with several state-of-the-art LLMs, including Claude, Gemini, DeepSeek, and Qwen. The experimental results (Table 14) indicate that the choice of LLM does not significantly influence projection performance

Table 12: Results of different base models with and w/o TTPL

| Method | TSP1K Obj.(Gap) | TSP5K Obj.(Gap) | TSP10K Obj.(Gap) |
|---|---|---|---|
| POMO greedy | 33.18 (43.51%) | 94.04 (84.50%) | 150.64 (109.86%) |
| POMO-TTPL | **29.68 (28.37%)** | **71.76 (40.79%)** | **101.8 (41.82%)** |
| SIGD greedy | **23.57(1.96%)** | 57.19 (12.20%) | 93.80 (30.68%) |
| SIGD-TTPL | 23.86 (3.20%) | **56.05 (9.97%)** | **80.44 (12.06%)** |
| BQ greedy | 23.65 (2.30%) | 58.27 (14.31%) | 89.73 (25.02%) |
| BQ-TTPL | **23.58 (1.99%)** | **56.47(10.79%)** | **81.48 (13.51%)** |
| LEHD greedy | 23.84 (3.11%) | 58.85 (15.46%) | 91.33 (27.24%) |
| LEHD-TTPL | **23.73 (2.65%)** | **52.63 (3.25%)** | **74.39 (3.63%)** |

Table 13: Time used for TTPL to learn projection strategies on different scales

| Mehod | TSP1K time | TSP5K time | TSP10K time |
|---|---|---|---|
| TTPL | 5.5h | 9.6h | 19.9h |

Table 14: Result of TTPL using different LLMs

| Method | TSP1K Obj. | TSP5K Obj. | TSP10K Obj. |
|---|---|---|---|
| TTPL-Claude | 23.80 | 52.79 | 74.48 |
| TTPL-DeepSeek | 23.67 | 52.39 | 74.00 |
| TTPL-Gemini | 23.76 | 53.71 | 76.42 |
| TTPL-Qwen | 23.75 | 52.73 | 74.25 |
| TTPL-GPT-4o-mini | 23.73 | 52.63 | 74.39 |

# F Solution Visualization

## F.1 Solution Visualizations of Cross-distribution TSP Instances

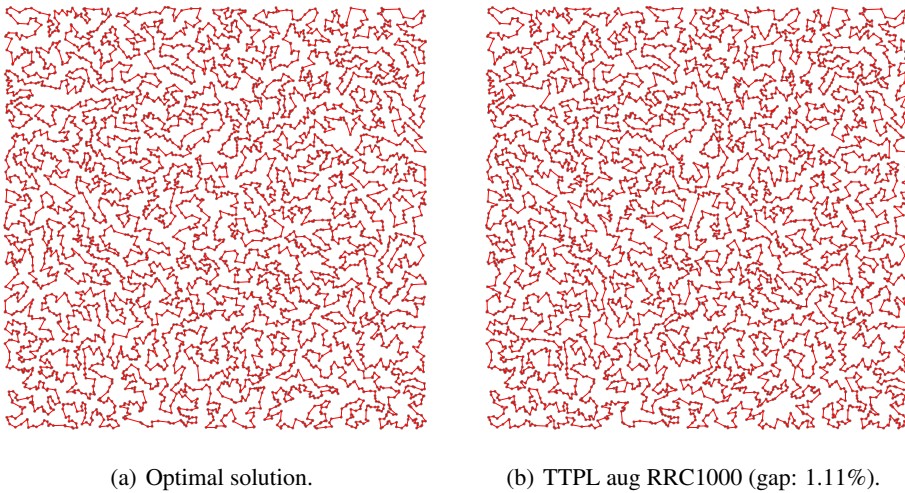

(a) Optimal solution.

(b) TTPL aug RRC1000 (gap: 1.11%).

Figure 3: The solution visualizations of a TSP5K instance with uniform distribution.

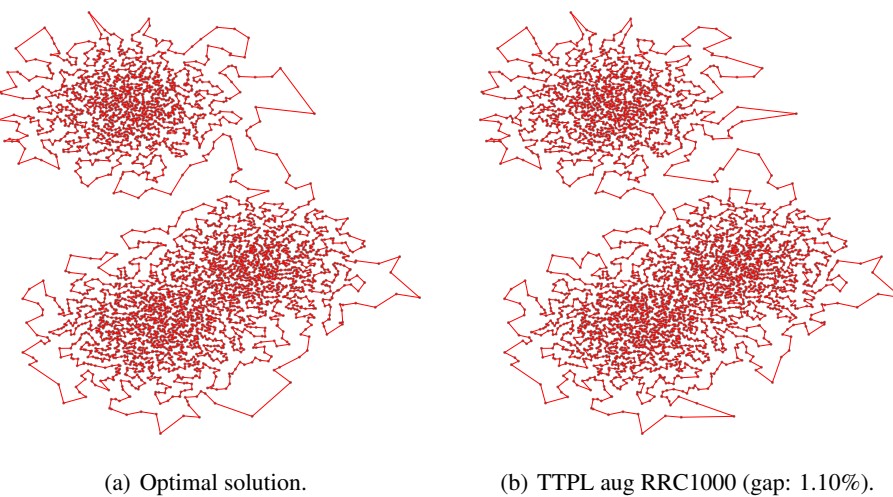

(a) Optimal solution.

(b) TTPL aug RRC1000 (gap: 1.10%).

Figure 4: The solution visualizations of a TSP5K instance with cluster distribution.

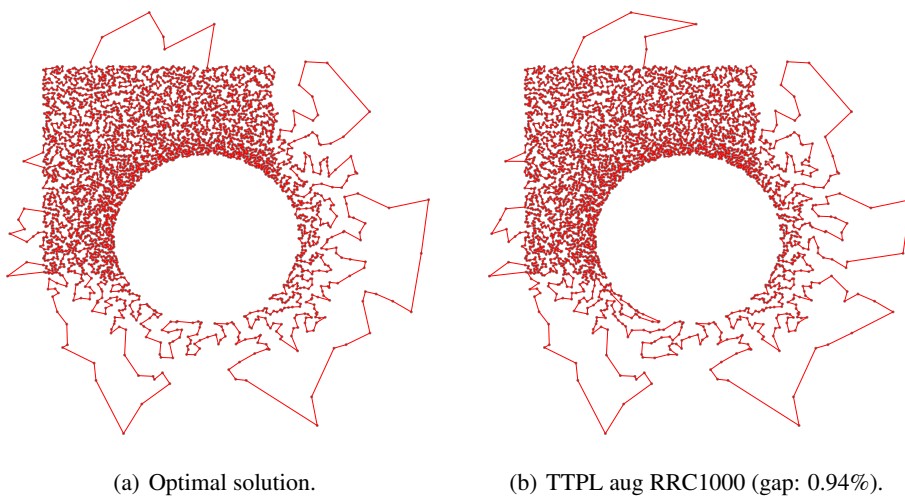

(a) Optimal solution.      (b) TTPL aug RRC1000 (gap: 0.94%).

Figure 5: The solution visualizations of a TSP5K instance with explosion distribution.

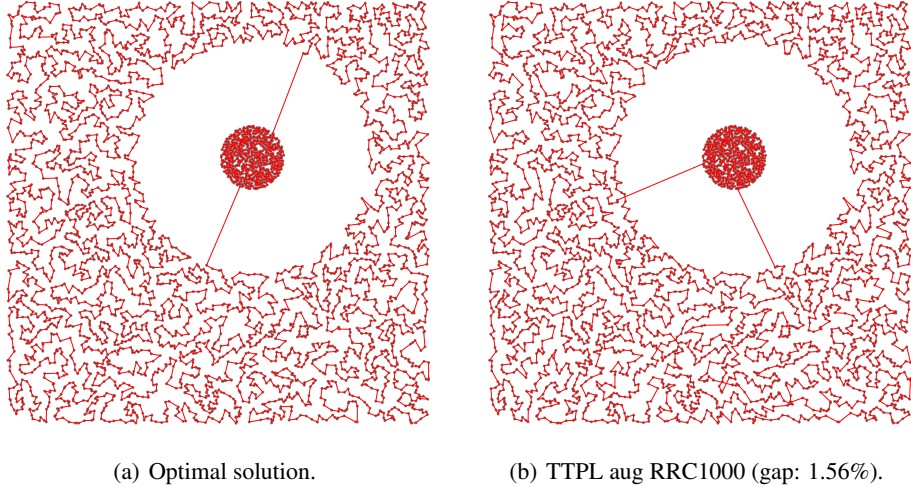

(a) Optimal solution.      (b) TTPL aug RRC1000 (gap: 1.56%).

Figure 6: The solution visualizations of a TSP5K instance with implosion distribution.

## F.2 Solution Visualizations of Large-scale TSPLIB Instances

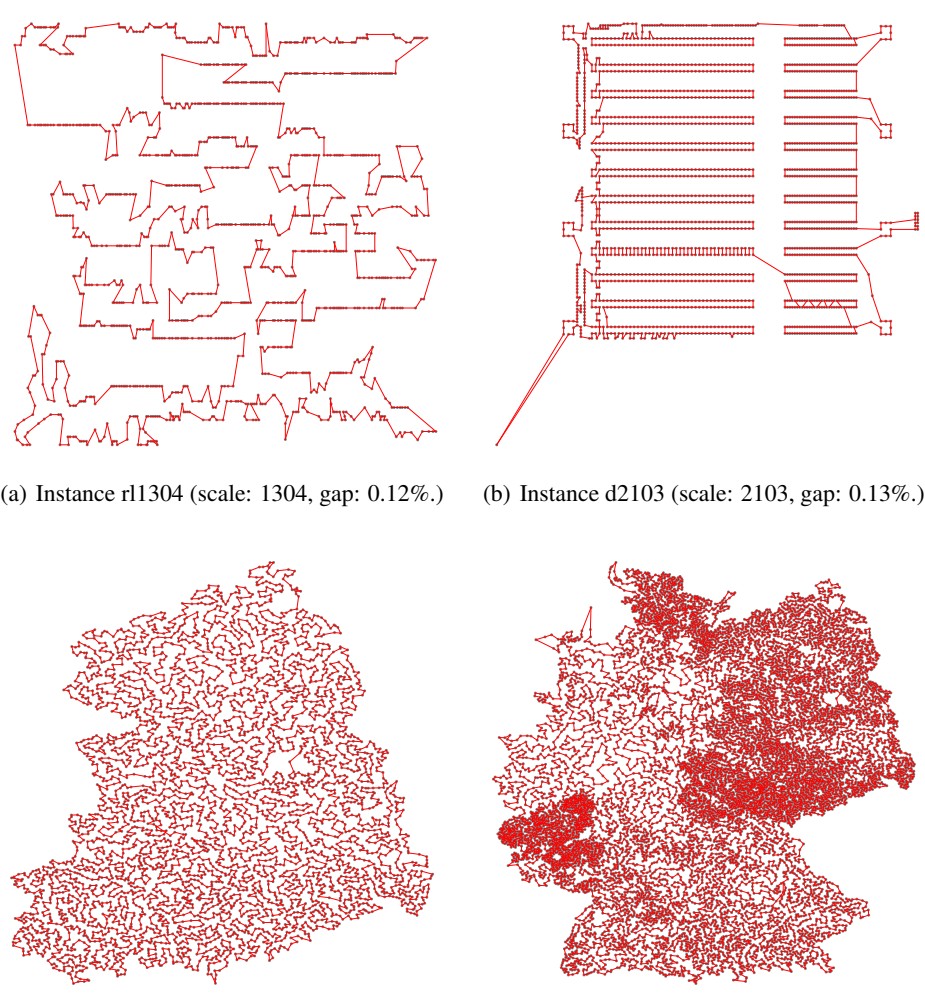

(a) Instance rl1304 (scale: 1304, gap: 0.12%.)    (b) Instance d2103 (scale: 2103, gap: 0.13%.)

(c) Instance fnl4461 (scale: 4461, gap: 1.10%.)    (d) Instance d15112 (scale: 15112, gap: 1.06%.)

Figure 7: The solution visualizations of TSPLIB[59] instances with different scales, the solutions are all generated by TTPL aug RRC1000.

# G License

We list the license of the code and the dataset in the Table. 15.

Table 15: A summary of licenses

| Resource | Type | Link | License |
|---|---|---|---|
| LKH3[57] | Code | http://webhotel4.ruc.dk/keld/research/LKH-3/ | Available for academic research use |
| HGS[58] | Code | https://github.com/chkwon/PyHygese | MIT License |
| Concorde[82] | Code | https://github.com/jvkersch/pyconcorde | BSD 3-Clause License |
| POMO[7] | Code | https://github.com/yd-kwon/POMO | MIT License |
| LEHD[18] | Code | https://github.com/CIAM-Group/NCO_code/tree/main/ single_objective/LEHD | MIT License |
| BQ[42] | Code | https://github.com/naver/bq-nco | CC BY-NC-SA 4.0 |
| GLOP[13] | Code | https://github.com/henry-yeh/GLOP | MIT License |
| H-TSP[12] | Code | https://github.com/Learning4Optimization-HUST/H-TSP | MIT License |
| DIFUSCO[28] | Code | https://github.com/Edward-Sun/DIFUSCO | MIT License |
| INViT[15] | Code | https://github.com/Kasumigaoka-Utaha/INViT | Available for academic research use |
| ELG[14] | Code | https://github.com/gaocrr/ELG | MIT License |
| SIGD[62] | Code | https://github.com/grimmlab/gumbeldore | Available for academic research use |
| EoH[52] | Code | https://github.com/FeiLiu36/EoH | MIT License |
| LLM4AD[83] | Code | https://github.com/Optima-CityU/LLM4AD | MIT License |
| TSPLIB[59] | Dataset | http://comopt.ifi.uni-heidelberg.de/software/TSPLIB95/ | Available for any non-commercial use |
| CVRPLib[60] | Dataset | http://vrp.galgos.inf.puc-rio.br/index.php/en/ | Available for academic research use |

# H Broader Impacts

This work advances neural combinatorial optimization through the proposed TTPL framework and MVFD technique, critically enhancing solution efficiency for large-scale Vehicle Routing Problems (VRPs). These developments are expected to inspire further investigations into novel neural methods for complex, large-scale routing tasks. We foresee no negative societal impacts from this research.

