# OpenReview forum: "Improving Generalization of Neural Combinatorial Optimization for Vehicle Routing Problems via Test-Time Projection Learning"
_NeurIPS.cc/2025/Conference — NeurIPS 2025 poster_

### Official Review · Reviewer_WCWc · 2025-06-12

**Clarity:** 3
**Significance:** 3
**Originality:** 2
**Rating:** 4
**Confidence:** 4

**Summary:**

This paper proposes a projection learning framework TTPL based on Large Language Model (LLM). By optimizing the projection strategy and multi-view decision fusion (MVDF) technology, the generalization ability of the neural combinatorial optimization model in large-scale vehicle routing problems (VRP) is significantly improved.

**Questions:**

Mainly see weaknesses.
Small issues:
1.What is the full name of AHD? Please give its full name when first appeared in the text.

**Ethical Concerns:**

["NO or VERY MINOR ethics concerns only"]

**Final Justification:**

The concerns have been addressed, however, consider the originality of the paper , I will keep my score.

**Limitations:**

Yes.

**Paper Formatting Concerns:**

No concerns about paper formatting.

**Quality:**

3

**Strengths And Weaknesses:**

Strengths:
1.After conducting sufficient ablation experiments and general experiments, I believe the MVDF you proposed has independent significance.
2.The algorithm in this paper successfully solves the problems of TSP-100K and CVRP-100K in an extremely short time, which shows the potential of LLM.
Weaknesses:
1.Although many comparative experiments have been conducted, as far as I know, this is not the first work to apply LLM to the solution of TSP problems. Perhaps, you should compare the results of your model with theirs. For example, the citation [46] in your article. Meanwhile, in the "Related Work" section, if possible, some introductions to the LLM-based generalization method should also be added.
2.You only used GPT-4o-mini and did not use other LLMS. The generalization of the method might be insufficient.
3.I don't know whether the LLM also uses your training data as the pre-training corpus. This will raise a question: Is the excellent performance of your model due to the good performance of the LLM itself or the effectiveness of the solution you proposed? This will greatly affect the assessment of innovation.

---

> ### Author Rebuttal · Authors · 2025-07-31
>
> Dear Reviewer WCWc
>
> Thank you very much for your time and effort in reviewing our work. We are glad to know you recognize the independent significance of MVDF and our method's efficiency in solving TSP/CVRP-100K problems. We address your concerns as follows.
>
>
> > **W1: Although many comparative experiments have been conducted, as far as I know, this is not the first work to apply LLM to the solution of TSP problems. Perhaps, you should compare the results of your model with theirs. For example, the citation [46] in your article. Meanwhile, in the "Related Work" section, if possible, some introductions to the LLM-based generalization method should also be added.**
>
> **A1:** We sincerely thank you for this insightful comment and for highlighting the important work in EoH [1]. We agree that a direct comparison and a more comprehensive discussion of related methods are crucial for claiming our contribution.
> **1. Comparison with EoH**
> In response to your suggestions, we have conducted a comparison experiment between our method and EoH on the TSP task. In particular, we run EoH on TSP1K/5K/10K. We use 128 instances for TSP1K and 16 instances for TSP5K and TSP10K. Moreover, we compare EoH's result with the solution obtained by TTPL using Random Re-Construction (RRC) under the same number of iterations. This is mainly because of the iterative nature of the EoH's implementation when solving TSP, which utilizes a Guided Local Search (GLS) algorithm to iteratively refine solutions. To ensure a fair comparison, we set the iterations of RRC and GLS to the same number. The final average solution length over test instances is reported in Table 17:
>
> Table 17: Comparison results on synthetic TSP dataset
> | Method       | TSP1K | TSP5K | TSP10K |
> |--------------|-------|-------|--------|
> | EoH-GLS1000  | 23.58 | 53.39 | 75.02  |
> | TTPL-RRC1000 | **23.19**  | **51.50** | **72.59**  |
>
> It is obvious that our proposed method consistently outperforms the EoH over all test scales. Specifically, TTPL reduces the solution length by 1.68\%, 3.67\%, and 3.35\% for the TSP1K, 5K, and 10K instances, respectively. These results indicate the superior performance of TTPL over LLM-based generalization methods.
>
> **2. Additional review for LLM-based generalization methods**
> As suggested, we will include a discussion of recent LLM-based generalization methods for combinatorial optimization in **Appendix A, page 14**. The details of the introduction to LLM-based generalization methods are as follows:
>
> Recent advancements have demonstrated the significant potential of Large Language Models (LLMs) in automating the design of high-performing heuristics for complex computational problems. This methodology typically involves maintaining a population of elite heuristic functions, which are evaluated and ranked based on their fitness on a designated dataset. The LLM is then iteratively prompted with high-performing functions from this population to generate potentially superior heuristic candidates. EoH [1] and Funsearch [2] pioneered applying LLM to design a population-based evolutionary computation procedure. Building on this, ReEvo [3] incorporated a reflection mechanism to augment the LLM's reasoning capabilities during heuristic generation. Addressing the challenge of premature convergence, HSEvo [4] introduced diversity metrics and a harmony search algorithm to enhance population diversity without degrading solution quality. In a different approach, MCTS-AHD [5] conceptualizes the task as a search problem, modeling the heuristic design process as a Monte Carlo Tree Search (MCTS) to systematically explore the solution space.
>
> We hope the above response can address your concerns.
>
> > **W2. You only used GPT-4o-mini and did not use other LLMs. The generalization of the method might be insufficient.**
>
> **A2:** We thank the reviewer for raising this important concern about the LLM's generalizability of our method. We agree that testing with a single LLM is a limitation because TTPL highly relies on the LLMs.
>
> To address this, we have conducted extensive experiments by integrating several state-of-the-art LLMs, including GPT4o-mini, Claude, Gemini, DeepSeek, and Qwen.
> We have employed all five LLMs to search for projection strategies on TSP1K instances and tested them on TSP1K, 5K, and 10K instances with LEHD as its base model. The results are summarized in Table 18.
>
> LEHD with the TTPL projection achieves better performance compared to its original version across different LLMs. In particular, TTPL helps LEHD reduce the solution length by 0.4, 3.79, and 7.88 of TSP1K, TSP5K, and TSP10K, respectively, with the **worst** projections, while shortening the solution length by 0.53, 5.11, and 10.3 of TSP1K, TSP5K, and TSP10K, respectively, with the **best** projections. It clearly demonstrates the generalization of the method in terms of LLM's usage.
>
> Table 18: Result of TTPL using different LLMs
> | Method           | TSP1K           | TSP5K           | TSP10K          |
> |------------------|-----------------|-----------------|-----------------|
> |                  | Obj. ($\Delta$) | Obj. ($\Delta$) | Obj. ($\Delta$) |
> | LEHD greedy      | 24.2 (0)        | 57.5 (0)        | 84.3 (0)        |
> | TTPL-Claude      | 23.80 (-0.4)    | 52.79 (-4.71)   | 74.48 (-9.82)   |
> | TTPL-DeepSeek    | **23.67 (-0.53)**   | **52.39 (-5.11)**   | **74.00 (-10.3)**  |
> | TTPL-Gemini      | 23.76 (-0.44)   | 53.71 (-3.79)   | 76.42 (-7.88)   |
> | TTPL-Qwen        | 23.75 (-0.45)   | 52.73 (-4.77)   | 74.25 (-10.05)  |
> | TTPL-GPT-4o-mini | 23.73 (-0.47)   | 52.63 (-4.87)   | 74.39 (-9.91)   |
>
> > **W3. I don't know whether the LLM also uses your training data as the pre-training corpus. This will raise a question: Is the excellent performance of your model due to the good performance of the LLM itself or the effectiveness of the solution you proposed? This will greatly affect the assessment of innovation.**
>
> **A3:** We sincerely apologize for the unclear description in our manuscript and thank you for your valuable feedback regarding LLMs' usage. We would like to reassure that the LLM utilized in this paper serves as **a zero-shot learner and was not fine-tuned or trained on any of the training datasets**. Consequently, the LLM had no prior exposure to the specific problem instances used in our experiment evaluation.
>
> Our innovation lies in the introduction of **projection learning** to improve the NCO methods' large-scale generalization capability. Different from the existing generalization methods that focus on model architecture or training paradigm, we propose a data-centric approach. It **only operates at the inference stage** by aligning the distribution of testing and training data.
>
> To implement the projection learning, heuristic-based methods demand significant domain expertise, while learning-based methods can be hindered by intricate feature engineering and the complex design of loss functions. Recently, LLM-based Automatic Heuristic Design (AHD) has emerged as a promising approach by leveraging the reasoning and code generation capabilities of LLMs to automatically discover and construct effective heuristics. In this paper, we utilize LLM-AHD methods as a design tool to search for a projection strategy.
>
> Nonetheless, we wish to clarify that a simple projection still can significantly improve the base model's performance. You may kindly refer to **Table 4 (page 8)**. In this experiment, we replaced our LLM-designed projection with a human-crafted strategy and applied it to the LEHD base model. The results demonstrate that even this basic projection leads to a significant performance improvement over the original LEHD version.
>
> This confirms that the proposed projection strategy methodology contributes most to the reported performance gains. The LLM functions as an enabling tool that automates the design of this projection, helping us achieve a robust and unified solution across multiple base models. We hope this clarification adequately addresses your concerns.
>
> > **Q1. What is the full name of AHD? Please give its full name when first appeared in the text.**
>
> **A4:** We sincerely thank you for pointing out this oversight. We apologize for failing to define the acronym "AHD" at its first mention. The full name for AHD is Automatic Heuristic Design.
>
> Thank you for your time and effort dedicated to reviewing our work. We sincerely hope that our responses have effectively addressed your concerns.
>
> **Reference**
>
> [1] Liu, F., Tong, X., Yuan, M., Lin, X., Luo, F., Wang, Z., ... & Zhang, Q. (2024). Evolution of Heuristics: Towards efficient automatic algorithm design using large language model. arXiv preprint arXiv:2401.02051.
>
> [2] Romera-Paredes, B., Barekatain, M., Novikov, A., Balog, M., Kumar, M. P., Dupont, E., ... & Fawzi, A. (2024). Mathematical discoveries from program search with large language models. Nature, 625(7995), 468-475.
>
> [3] Ye, H., Wang, J., Cao, Z., Berto, F., Hua, C., Kim, H., ... & Song, G. (2024). ReEvo: Large language models as hyper-heuristics with reflective evolution. Advances in neural information processing systems, 37, 43571-43608.
>
> [4] Dat, P. V. T., Doan, L., & Binh, H. T. T. (2025, April). HsEvo: Elevating automatic heuristic design with diversity-driven harmony search and genetic algorithm using llms. In Proceedings of the AAAI Conference on Artificial Intelligence (Vol. 39, No. 25, pp. 26931-26938).
>
> [5] Zheng, Z., Xie, Z., Wang, Z., & Hooi, B. (2025). Monte carlo tree search for comprehensive exploration in llm-based automatic heuristic design. arXiv preprint arXiv:2501.08603.

---

> > ### Comment · Reviewer_SjSP · 2025-08-05
> >
> > Thank you for the authors’ response, which has clarified some of my concerns. I acknowledge that the proposed approach presents a good application of the EoH framework to VRPs. I still have reservations regarding the long computation time of the LLM.

---

> > > ### Author Response · Authors · 2025-08-09
> > >
> > > Dear Reviewer WCWc \&  Reviewer SjSP,
> > >
> > > We have addressed the concerns in the response to reviewer SjSP, entitled "Response regarding computation time of the LLM".

---

### Official Review · Reviewer_3Y2f · 2025-06-30

**Clarity:** 3
**Significance:** 4
**Originality:** 3
**Rating:** 4
**Confidence:** 4

**Summary:**

This paper addresses the significant challenge of generalization in Neural Combinatorial Optimization (NCO) for Vehicle Routing Problems (VRPs). While NCO models perform well on problem instances of similar scale to their training data (typically small instances like 100 nodes), their performance degrades significantly when applied to larger-scale problems due to a distributional shift between training and testing data.

To overcome this limitation, the paper introduces a novel learning framework called Test-Time Projection Learning (TTPL). This framework is designed to learn a projection between the training and testing data distributions, which is then used to enhance the scalability and generalization of existing NCO models.

The key contributions of this work are:

A novel learning framework driven by Large Language Models (LLMs): This framework learns a projection that bridges the distributional gap between small-scale training data and large-scale test data.

Inference-only approach: Unlike many existing techniques that require joint training with the neural network, TTPL operates exclusively during the inference phase. This means it can be applied to pre-trained NCO models without needing to retrain them, making it highly flexible and practical.

Enhanced scalability for NCO models: Extensive experiments demonstrate that TTPL significantly improves the performance of backbone NCO models (trained on 100-node instances) when applied to larger-scale Traveling Salesman Problems (TSP) and Capacitated Vehicle Routing Problems (CVRP) up to 1000 nodes,

**Questions:**

Question: The paper states that LLMs drive the learning of projection strategies. Could the authors provide a more in-depth explanation of how the LLM specifically learns this projection? What kind of prompts are used? What is the input/output format for the LLM during this process? Is the LLM fine-tuned, or is it used as a zero-shot/few-shot learner? What is the architecture of this "projection learning" component?
A clear and comprehensive explanation of the LLM's internal mechanism for projection learning would significantly improve the Quality and Clarity ratings.

Question: While TTPL avoids retraining the NCO backbone, it introduces an additional "test-time projection learning" step. Could the authors provide a more detailed breakdown of the computational overhead (inference time and memory) introduced by the TTPL framework itself, distinct from the backbone NCO model's inference? This is particularly relevant for large-scale instances (e.g., N=1000).
Suggestion: Include a table or graph comparing the inference time of the backbone model alone versus the backbone + TTPL, perhaps also breaking down the TTPL component's time. This would offer a more complete picture of the practical efficiency trade-offs.

Question: The method shows strong generalization on TSP and CVRP to larger scales. How do the authors anticipate TTPL performing on other, more complex Combinatorial Optimization Problems (COPs) or more constrained VRP variants (e.g., VRP with time windows, multiple depots, pick-up and delivery, heterogeneous fleets)?
Suggestion: Discuss the potential applicability and limitations of TTPL for a broader range of COPs. A brief qualitative discussion on how the framework might need to be adapted (e.g., new feature representations, different projection strategies) for problems with more complex constraints would be valuable.

Question: Are there specific parameters or hyper-parameters within the projection learning component (e.g., related to the LLM's usage, projection dimension, or learning rate for the projection) that significantly influence TTPL's performance? How robust is TTPL to variations in these parameters?

**Ethical Concerns:**

["NO or VERY MINOR ethics concerns only"]

**Limitations:**

Yes.

**Paper Formatting Concerns:**

No.

**Quality:**

3

**Strengths And Weaknesses:**

Strengths:

Quality: The paper presents a novel and well-motivated approach to address a crucial problem in NCO: generalization to larger problem scales. The proposed Test-Time Projection Learning (TTPL) framework is innovative, particularly its inference-only nature. The experimental evaluation is comprehensive, testing TTPL on both TSP and CVRP benchmarks across various scales (from 100-node training to 1000-node testing). The results clearly demonstrate the effectiveness of TTPL in improving the generalization ability of backbone NCO models, outperforming strong baselines.

Clarity: The paper is well-written and easy to understand. The problem of distributional shift and generalization in NCO is clearly articulated. The TTPL framework, including the role of LLMs and the projection learning mechanism, is explained logically. Figures (e.g., Figure 1 illustrating the framework) are helpful in conveying complex ideas. The experimental setup, including baseline models and evaluation metrics, is clearly described.

Significance: The problem of generalization across different scales is a major hurdle for the practical deployment of NCO models. By proposing an effective inference-only solution, this paper makes a highly significant contribution. TTPL's ability to enhance pre-trained models without retraining is a practical advantage that can save significant computational resources and make NCO more accessible for real-world large-scale problems. The use of LLMs to drive this projection learning is also significant, exploring a novel application area for LLMs in optimization.

Originality: The core idea of "test-time projection learning" to address generalization in NCO is highly original. The framework's unique characteristic of operating solely during inference to bridge the distributional gap, and the novel integration of LLMs to learn this projection, are innovative contributions that distinguish this work from previous efforts in NCO generalization (which often involve data augmentation or architectural changes during training).

Weaknesses:

LLM Role and Transparency: While the paper states LLMs are used to drive the learning of projection strategies, the exact mechanism and specific LLM (or family of LLMs) used for this "projection learning" could be more transparently detailed. The justification for LLM usage in the declaration section ("designing projection strategies") is somewhat vague. More specifics on the LLM's architecture within the framework, its training (if any specific to TTPL), and how it concretely learns and applies the projection would strengthen this aspect.

Theoretical Guarantees/Insights: The paper primarily focuses on empirical results. While impressive, a deeper theoretical analysis into why TTPL works, or under what conditions the projection learning is guaranteed to succeed, would significantly strengthen the foundational understanding. For instance, what properties of the LLM or the projection mechanism enable it to effectively bridge distributional shifts?

Computational Overhead of LLM/TTPL: While TTPL avoids NCO model retraining, the inference-time projection learning itself might introduce a computational overhead. A more detailed analysis of the time and resource costs associated with the LLM-driven projection learning component during inference, especially for very large instances, would be beneficial for practical considerations.

Generalization to Other Combinatorial Problems: The method is evaluated on TSP and CVRP. While these are fundamental VRPs, discussing how TTPL might extend to other combinatorial optimization problems (e.g., Orienteering Problem, Prize Collecting TSP, other routing problems with complex constraints) would broaden its perceived applicability.

---

> ### Author Rebuttal · Authors · 2025-07-31
>
> Dear Reviewer 3Y2f
>
> Thank you very much for your time and effort in reviewing our work. We are glad to know you find TTPL novel and well-motivated, and recognize its significance for practical NCO deployment. Your concerns are answered as follows.
>
> > **W1 & Q1. LLM Role and Transparency: The paper's explanation of how the Large Language Model (LLM) learns "projection strategies" is too vague and lacks transparency. To improve the quality and clarity of the work, please provide a more detailed description of this mechanism, specifying the architecture of the projection learning component, the prompts and input/output formats used, and whether the LLM is fine-tuned or utilized in a zero-shot or few-shot capacity.**
>
> **A1:** Thanks very much for raising concerns about our methodology. The LLM in TTPL framework undergoes **no training or fine-tuning**. During TTPL iteration, LLM generates and refines **executable code** for the projection strategies.
>
> Specifically, the input is a crafted natural language prompt, which includes the **task description, evolution instructions, and template program**. Task description helps LLM to understand the problem, providing prior information. Evolution instructions indicate the LLM to do reasoning over the in-context information and generate new projection strategies as well as their corresponding code implementations. The template program serves as the seed to guide the optimization direction of LLM. You may kindly refer to **Appendix B.1 (page 14-18)** for more details about prompts.
>
> The output is an executable source code (e.g., a Python function) that implements a projection strategy. It should be noted that any existing LLMs (e.g., models from the GPT series, Claude, Gemini, etc.) could be integrated into TTPL.
>
> > **W2. Theoretical Guarantees/Insights: The paper primarily focuses on empirical results. While impressive, a deeper theoretical analysis into why TTPL works, or under what conditions the projection learning is guaranteed to succeed, would significantly strengthen the foundational understanding. For instance, what properties of the LLM or the projection mechanism enable it to effectively bridge distributional shifts?**
>
> **A2**: Thank you for your professional comments. We fully agree that providing a deeper analysis of why our projection learning strategy is effective is critical to understanding the significance of this work.
>
> To provide a theoretical foundation for the effectiveness of our projection strategy, we employ the Rigid Wasserstein Distance (RWD) [1] as a metric. **RWD quantifies the similarity between the K-Nearest-Neighbor (KNN) subgraph extracted from the test instance and the target training instance**. Specifically, RWD models CO instances (TSP/CVRP in our study) as probability measures, and the RWD calculates the minimum cost to make them identical under optimal transformations. The formal definition can be written as
> $$
>     V(\mu, v) := (\underset{e \in E(d)}{\mathrm{min}}\sum_{i=1}^n \sum_{j=1}^nC_{ij}||x_i - e(y_j)||^2)^{\frac{1}{2}},
> $$
> where $\mu$ and $v$ are the probability measures of two different CO instances, $C$ is the cost matrix $C_{ij} = ||x_i - y_i||$ represents distance that transport point $x_i$ to $y_i$, $e \in E(d)$ indicates rigid transformations (translation, rotation, reflection, etc.), $||x_i - e(y_j)||$ is the distance between a point $x_i$ from the first instance and a transformed point $e(y_i)$ from the second instance. It is computed after an optimal alignment transformation $e$ has been applied.
>
> To empirically demonstrate how our learned projection strategy improves performance by mitigating the distributional shift between training and test data, we analyzed the KNN subgraphs generated by the LEHD model on a TSP1K instance. For a given training instance, we computed the RWD between it and the KNN subgraphs, both before and after applying our TTPL projection. We then correlated these RWD values with the final solution quality achieved by the LEHD model. The results of this analysis are presented in Table 12.
>
> Table 12: Illustration of distribution difference and performance
> |Method|RWD|Obj.|
> |-|-|-|
> |LEHD|0.121|57.50|
> |LEHD-TTPL|**0.026**|**52.63**|
>
> The data in the table reveals a strong correlation: a lower RWD in the projected KNN subgraphs corresponds directly to a higher quality solution. This suggests that the model's predictive performance is enhanced when the input graph's distribution is more similar to that of the training instances.
>
> > **W3. Computational Overhead of LLM/TTPL: While TTPL avoids NCO model retraining, the inference-time projection learning itself might introduce a computational overhead. A more detailed analysis of the time and resource costs associated with the LLM-driven projection learning component during inference, especially for very large instances, would be beneficial for practical considerations.**
> > **Q2. While TTPL avoids retraining the NCO backbone, it introduces an additional "test-time projection learning" step. Could the authors provide a more detailed breakdown of the computational overhead (inference time and memory) introduced by the TTPL framework itself, distinct from the backbone NCO model's inference? This is particularly relevant for large-scale instances (e.g., N=1000). Suggestion: Include a table or graph comparing the inference time of the backbone model alone versus the backbone + TTPL, perhaps also breaking down the TTPL component's time. This would offer a more complete picture of the practical efficiency trade-offs.**
>
> **A3:** We sincerely thank you for raising this important point regarding the computational overhead of our TTPL framework. Following your advice, we have performed a detailed breakdown of the computation cost introduced by TTPL on TSP1K/5K/10K. The time is listed in Table 13.
>
> Table 13: Time usage breakdown for TTPL on TSP instances
> |Problem Size|Inference|Inference|Learning|
> |-|-|-|-|
> ||Backbone|Backbone+TTPL|TTPL|
> |1000|0.1s|0.1s|5.5h|
> |5000|1.4s|1.4s|9.6h|
> |10000|2.7s|2.9s|19.9h|
>
> The TTPL learning process, which involves the LLM-driven optimization, is indeed computationally intensive, potentially taking several hours. However, we would like to emphasize that this is **a one-time, offline cost**. In particular, once a projection strategy is discovered, it is highly **generalizable and imposes negligible computational overhead during inference**. Therefore, the initial time investment is worthwhile, as TTPL consistently improves the performance of base models on large-scale instances.
>
> > **W4. Generalization to Other Combinatorial Problems: The method is evaluated on TSP and CVRP. While these are fundamental VRPs, discussing how TTPL might extend to other combinatorial optimization problems (e.g., Orienteering Problem, Prize Collecting TSP, other routing problems with complex constraints) would broaden its perceived applicability.**
> > **Q3. The method shows strong generalization on TSP and CVRP to larger scales. How do the authors anticipate TTPL performing on other, more complex Combinatorial Optimization Problems (COPs) or more constrained VRP variants (e.g., VRP with time windows, multiple depots, pick-up and delivery, heterogeneous fleets)? Suggestion: Discuss the potential applicability and limitations of TTPL for a broader range of COPs. A brief qualitative discussion on how the framework might need to be adapted (e.g., new feature representations, different projection strategies) for problems with more complex constraints would be valuable.**
>
> **A4:** We appreciate your constructive feedback. To demonstrate the generalizability of our method, we have acted on your suggestion and extended our experiments. Specifically, we integrated our TTPL projection with the Sym-NCO [2] model and successfully applied it to the Orienteering Problem (OP). This confirms that the proposed TTPL projection is not confined to a specific problem architecture. Although the present implementation is coordinate-based and potentially limited when coordinate information is not available, we will extend it to a distance-based projection, broaden its applicability to more complex VRPs, such as the Asymmetric TSP.
>
>
> Table 14: TTPL with Sym-NCO on OP instances
> |Method|OP1K|OP5K|OP10K|
> |-|-|-|-|
> |Sym-NCO|89.92|239.40|428.22|
>
> > **Q4. Are there specific parameters or hyper-parameters within the projection learning component (e.g., related to the LLM's usage, projection dimension, or learning rate for the projection) that significantly influence TTPL's performance? How robust is TTPL to variations in these parameters?**
>
> **A5:** We thank you for this insightful question about TTPL's robustness on its hyperparameters. To investigate this, we have conducted a series of ablation studies focusing on the **population size** and the **choice of the LLM**. We report results in Tables 15 and 16, demonstrating that the framework's performance remains stable across these variations. This confirms that TTPL is robust with respect to these
>
> Table 15: Result of TTPL using different population sizes
> |Population size|10|15|20|25|30|
> |-|-|-|-|-|-|
> |TTPL|23.68|23.80|23.73|23.68|23.75|
>
> Table 16: Result of TTPL using different LLMs
> |Method|TSP1K|TSP5K|TSP10K|
> |-|-|-|-|
> |TTPL-Claude|23.80|52.79|74.48|
> |TTPL-DeepSeek|23.67|52.39|74.00|
> |TTPL-Gemini|23.76|53.71|76.42|
> |TTPL-Qwen|23.75|52.73|74.25|
> |TTPL-GPT-4o-mini|23.73|52.63|74.39|
>
> Thanks again for your time and effort in reviewing our work. We sincerely hope that our responses have effectively addressed your concerns.
>
> **Reference**
> [1] Wang, X., Miao, F., Liu, W., \& Xiong, Y. (2025, March). Efficient and robust neural combinatorial optimization via Wasserstein-based coresets. In The Thirteenth International Conference on Learning Representations.
>
> [2] Kim, M., Park, J., \& Park, J. (2022). Sym-NCO: Leveraging symmetricity for neural combinatorial optimization. Advances in Neural Information Processing Systems, 35, 1936-1949.

---

### Official Review · Reviewer_SjSP · 2025-07-01

**Clarity:** 2
**Significance:** 3
**Originality:** 3
**Rating:** 4
**Confidence:** 4

**Summary:**

The paper proposes Test-Time Projection Learning (TTPL), an LLM-driven evolutionary framework that automatically designs geometric projection functions to reconcile the distribution gap between small-scale training instances (≈100 nodes) and large-scale inference instances (up to 100K nodes) in Vehicle Routing Problems (VRPs). A complementary Multi-View Decision Fusion (MVDF) module ensembles multiple transformed views of each K-nearest-neighbour sub-graph to further stabilise node selection. According to the manuscript, the combined TTPL-MVDF system reduces optimality gaps on large-scale Traveling Salesman Problem (TSP) and Capacitated VRP (CVRP) benchmarks by 60~88% relative to the best competing constructive methods.

**Questions:**

1. It is recommended to provide a more detailed theoretical explanation or interpretability experiment of why the proposed projection strategy improves model generalization in large-scale VRPs.
2. It is suggested to report mean ±σ over multiple random seeds or instance batches, and add paired-t confidence intervals for key gaps.
3. The loss functions and the evaluation metrics (e.g., RWD) used in this work needs to be discussed in detail in the main text.
4. It could be helpful if the author(s) explore ways to speed up the convergence of the LLM optimization phase to improve the practical applicability of TTPL in time-sensitive settings.
5. Can the authors elaborate on how their LLM-driven method compares to other projection methods (e.g., INViT, LEHD) in terms of computational efficiency, especially for real-time large-scale problems?
6. Given that TTPL relies on prompt-based LLM outputs, have the authors evaluated the consistency of generated strategies across different LLMs, prompts, or random seeds? Understanding this would help assess the method’s reproducibility and reliability.
7. Could the authors explain whether the framework could be generalized to other domains beyond VRPs, such as other NP-hard problems?

**Ethical Concerns:**

["NO or VERY MINOR ethics concerns only"]

**Final Justification:**

The proposed method is empirically effective, but increases computation time. I would keep my original score. I am OK if the paper is accepted.

**Limitations:**

Yes

**Quality:**

2

**Strengths And Weaknesses:**

Strength:
1. (Quality) The manuscript includes abundant experimental workloads. It evaluates extensive synthetic and real-world data across five instance sizes per problem. And the ablations isolate the benefit of TTPL and MVDF individually.
2. (Significance) The introduction of TTPL is a strong advancement in neural combinatorial optimization, improving model generalization on large-scale VRPs without requiring retraining, making it practical for real-world applications.
3. (Originality) The application of LLMs to design the projection strategies during the inference phase is a novel and promising approach, which is a unique contribution to the field.

Weaknesses:
1. (Quality) Lack of Theoretical Insights/Interpretability: The paper focuses primarily on experimental results and does not provide any deep theoretical analysis of why the projection strategy works or how it guarantees better performance.
2. (Quality) Lack of Statistical Significance: The paper does not provide error bars or other statistical measures to indicate the variability or reliability of the results, which would be crucial for validating the robustness of the method.
3. (Clarity) Some key information is missed: Some critical loss functions and evaluation metrics should be further discussed in the main text.
4. (Quality) Although results are promising, the paper does not fully address the computational cost, especially in terms of the time required for the LLM optimization process.
5. (Significance) The paper mentions the slow convergence of the LLM optimization phase, which is a significant practical limitation for deploying this method in real-time applications.
6. (Clarity) Several typos (e.g., “MVFD” spelled “MVDF”).

---

> ### Author Rebuttal · Authors · 2025-07-31
>
> Dear Reviewer SjSP
>
> Thank you very much for your time and effort in reviewing our work. We are more than delighted to know you find our method is novel and applicable in real-world scenarios. We address your concerns as follows.
>
> > **W1 & Q1: A deeper theoretical analysis or an interpretability experiment to explain why the proposed projection strategy improves the model's generalization performance.**
>
> **A1**: Thank you very much for raising this concern. We fully agree that providing a deeper analysis of why our projection learning strategy is critical to understanding the significance of this work.
>
> To theoretically explain the effectiveness of projection, we introduce Rigid Wasserstein Distance (RWD) [1] as a metric to **quantify the similarity between the K-Nearest-Neighbor (KNN) subgraph extracted from the test instance and the target training instance**. Specifically, RWD models CO instances (TSP/CVRP in our study) as probability measures, and the RWD calculates the minimum cost to make them identical under optimal transformations. The formal definition can be written as
> $$
>     V(\mu, v) := (\underset{e \in E(d)}{\mathrm{min}}\sum_{i=1}^n \sum_{j=1}^nC_{ij}||x_i - e(y_j)||^2)^{\frac{1}{2}},
> $$
> where $\mu$ and $v$ are the probability measures of two different CO instances, $C$ is the cost matrix $C_{ij} = ||x_i - y_i||$ represents distance that transport point $x_i$ to $y_i$, $e \in E(d)$ indicates rigid transformations, $||x_i - e(y_j)||$ is the distance between a point $x_i$ from the first instance and a transformed point $e(y_i)$ from the second instance.
>
> To empirically demonstrate how our learned projection strategy improves performance by mitigating the distributional shift between training and test data, we computed the RWD between a given training instance and the KNN subgraphs generated during the search process, both before and after applying our TTPL projection. We then correlated these RWD values with the final solution quality achieved by the LEHD model. The results of this analysis are presented in Table 4.
>
> Table 4: Illustration of distribution difference and performance
> |Method|RWD|Obj.|
> |-|-|-|
> |LEHD|0.121|57.50|
> |LEHD-TTPL|**0.026**|**52.63**|
>
> The data in Table 4 reveals that a lower RWD in the projected KNN subgraphs corresponds directly to a higher quality solution. This suggests that the model's predictive performance is enhanced when the input graph's distribution is more similar to that of the training instances.
>
> > **W2 & Q2: An inclusion of a statistical significance measure to validate the robustness and reliability of the method**
>
> **A2:** We sincerely appreciate your constructive comments. In response, we have addressed the statistical analysis in two stages of our methodology:
> **TTPL Search Process**
> To verify the stability of the TTPL search process, we perform 10 independent runs, each initialized with the same base model (LEHD) and settings. We then calculated the mean and standard deviation of the optimal gap of the final strategies obtained on TSP1K/5K/10K. The results are shown in Table 5:
>
> Table 5: Results of TTPL with 10 independent runs
> |Method|TSP1K|TSP5K|TSP10K|
> |-|-|-|-|
> ||mean$\pm$std.|mean$\pm$std.|mean$\pm$std.|
> |TTPL|23.77$\pm$3.22e-03|52.71$\pm$2.17e-02|74.36$\pm$2.06e-02|
>
> **Inference stage**
>
> To assess the performance of the projection found by TTPL, we evaluated it across different batches and calculated the mean and standard deviation as suggested. We have also proceeded with paired t-tests to formally assess the statistical significance of the solution quality between our methods and several strong baselines. The symbols "+", "=", and "-" denote that TTPL performs statistically better than, competitive with, and worse than the compared method, respectively.
>
> Table 6: Statistical results on TSP1K/5K/10K
> |Problem Size|INViT|BQ|LEHD|TTPL|
> |-|-|-|-|-|
> |TSP1K|24.66±0.30+|23.65±0.28=|23.84±0.34+|**23.64±0.36**|
> |TSP5K|54.49±0.14+|58.27±0.97+|58.85±0.38+|**52.43±0.30**|
> |TSP10K|76.85±0.20+|89.73±1.22+|91.33±1.33+|**73.49±0.44**|
> |+/=/-|3/0/0|2/1/0|3/0/0|NA|
>
> > **W3 & Q3: Detailed discussion of the key loss functions and evaluation metrics (e.g., RWD) in the main text**
>
> **A3:** We sincerely thank you for the feedback and apologize for the ambiguity in the description of our methodology. We will add the corresponding explanation to our main text in the revised version. The loss function and the evaluation metrics are described as follows:
> **Loss function**
> The backbone model used in this paper is LEHD, which is trained with the **Cross-Entropy** loss. It should be noted that the proposed TTPL framework **is strictly inference-only** and the LEHD backbone model is not trained or fine-tuned during the TTPL searching process.
> **Evaluation metrics**
> Fitness value: To guide the LLM in designing projection strategies, we use a fitness value defined as the solution length solved by the base model on a set of validation instances. This score measures the quality of a given strategy.
> Rigid Wasserstein Distance (RWD): In this paper, we use RWD as the metric to measure the difference between Combinatorial Optimization (CO) instances. The detailed explanation of RWD can be found in the previous answer (**A1**). In general, RWD is used to describe the distance between different CO instances. It first models CO instances as probability measures and then calculates the minimum cost required to transport one measure to the other.
>
> > **W4 & Q5: Detailed analysis of the method's computational cost and efficiency**
>
> **A4:** Thank you for your professional comments. We agree that showing the computational cost introduced by TTPL is important.
> In our main text, we adopt TTPL to search for the appropriate projection strategy on TSP1K, TSP5K, and TSP10K instances. To exhibit the computational cost, we have recorded the additional time introduced by the TTPL search process in Table 7.
>
> Table 7: Time used for TTPL to learn projection strategies on different scale
> |Method|TSP1K|TSP5K|TSP10K|
> |-|-|-|-|
> |TTPL|5.5h|9.6h|19.9h|
>
> Although TTPL introduces additional search costs, its primary contribution is the automated discovery of the best-fitted projection strategy. This automation contributes to TTPL's significant versatility, allowing it to generalize effectively across different base models rather than being confined to a single architecture.
>
>
> > **W5 & Q4: Exploring approaches to accelerate the convergence of the LLM optimization to improve the method's practical applicability**
>
> **A5:** We sincerely thank you for this insightful feedback to explore acceleration techniques. In response, we have implemented a parallel search strategy to enhance the efficiency of the optimization phase.
>
> To demonstrate the impact of this improvement, we conducted experiments to compare the time consumption of the optimization process with and without the parallel search strategy. The times are summarized in Table 8.
>
> Table 8: Results of TTPL using different search strategies
> |Method|TSP1K|TSP5K|TSP10K|
> |-|-|-|-|
> |TTPL (Sequential)|5.5h|9.6h|19.9h|
> |TTPL (Parallel)|**3.2h**|**5.6h**|**11.6h**|
>
> > **W6. Several typos (e.g., “MVFD” spelled “MVDF”).**
> **A6:** We sincerely apologize for the typo and appreciate you bringing it to our attention. We will correct them.
>
> > **Q6. Consistency and reliability of TTPL's outputs across different LLMs, prompts, or random seeds**
>
> **A7:** We thank you for this constructive question about the sensitivity of TTPL to its hyperparameter. To investigate this, we first tested our TTPL framework with several state-of-the-art LLMs, including Claude, Gemini, DeepSeek, and Qwen. We show the experimental results in Table 9. It is obvious that the choice of LLM has no significant influence on the projection performance.
>
> Table 9: Result of TTPL using different LLMs
> |Method|TSP1K|TSP5K|TSP10K|
> |-|-|-|-|
> |TTPL-Claude|23.80|52.79|74.48|
> |TTPL-DeepSeek|**23.67**|**52.39**|**74.00**|
> |TTPL-Gemini|23.76|53.71|76.42|
> |TTPL-Qwen|23.75|52.73|74.25|
> |TTPL-GPT-4o-mini|23.73|52.63|74.39|
>
> Besides, we examine the influence of different prompt strategies. In this paper, we provide a template program for the LLM to provide guidance in generating a new projection strategy. To examine TTPL's robustness, we remove this template program and let the LLM search on its own capabilities. We equipped LEHD with the obtained projection and tested on TSP1K/5K/10K instances. The results are shown in Table 10.
>
> Table 10: TTPL using different prompts
> |Method|TSP1K|TSP5K|TSP10K|
> |-|-|-|-|
> |LEHD w/o TTPL|24.2|57.5|84.3|
> |TTPL w/o template|24.00|57.13|82.77|
> |TTPL full prompt|**23.73**|**52.63**|**74.39**|
>
> > **Q7. Could the authors explain whether the framework could be generalized to other domains beyond VRPs, such as other NP-hard problems?**
>
> **A8:** Thank you for this insightful question. We sincerely apologize that our current implementation can not be generalized to other domains beyond VRPs. This is because the data formats and problem structures differ significantly across CO domains, and our current strategy is tailored to the coordinate-based routing problem.
>
> However, we generalized our method to the Orienteering Problem (OP). Specifically, we employed the TTPL in the Sym-NCO [2] model and tested it on large-scale OP instances. The results are summarized below:
>
> Table 11: Sym-NCO using TTPL on OP instances
> |Method|OP1K|OP5K|OP10K|
> |-|-|-|-|
> |Sym-NCO|89.92|239.40|428.22|
>
> Thank you for your time in reviewing our work. We hope that our responses have effectively addressed your concerns.
>
> **Reference**
>
> [1] Wang, X., Miao, F., Liu, W., \& Xiong, Y. (2025, March). Efficient and robust neural combinatorial optimization via Wasserstein-based coresets. In The Thirteenth International Conference on Learning Representations.
>
> [2] Kim, M., Park, J., \& Park, J. (2022). Sym-NCO: Leveraging symmetricity for neural combinatorial optimization. Advances in Neural Information Processing Systems, 35, 1936-1949.

---

> ### Author Response · Authors · 2025-08-08
> **Response regarding computation time of the LLM**
>
> Dear Reviewer SjSP
>
> We thank you for your continued engagement in the discussion. We appreciate the opportunity to address concerns regarding the computation time of the LLM. We address your concerns as follows:
>
> > **1. One-time learning process**
>
> We would like to clarify that the TTPL projection learning is a **one-time process**, where a projection developed for a specific problem type (e.g., TSP, CVRP) can be **applied to improve an NCO model's performance across instances of that problem**. To validate this, we transplanted the projection, which was initially designed based on LEHD, directly onto the POMO and BQ. We then evaluated their performance on the TSP1K-10K instances, with results presented in Table A1. From these results, we can observe that the model equipped with TTPL outperforms its original version on all test instances, confirming the adaptability of the proposed method across different models.
>
> Table. A1: Generalization of the LEHD-learned projection to other models
> |Method|TSP1K|TSP5K|TSP10K|
> |-|-|-|-|
> |POMO greedy|33.18(43.51\%)|94.04(84.50\%)|150.64(109.86\%)|
> |POMO-TTPL|**29.68(28.37\%)**|**71.76(40.79\%)**|**101.8(41.82\%)**|
> ||||
> |BQ greedy|23.65(2.30\%)|58.27(14.31\%)|89.73(25.02\%)|
> |BQ-TTPL|**23.58(1.99\%)**|**56.47(10.79\%)**|**81.48(13.51\%)**|
> ||||
> |LEHD greedy|23.84(3.11\%)|58.85(15.46\%)|91.33(27.24\%)|
> |LEHD-TTPL|**23.73(2.65\%)**|**52.63(3.25\%)**|**74.39(3.63\%)**|
>
> > **2. Computation time of the LLM**
>
> We acknowledge that employing TTPL during the inference phase incurs an additional computational overhead of several hours. However, this temporal investment significantly enhances the model's generalization capabilities on large-scale instances. To empirically evaluate its cost-effectiveness, we quantified the computational overhead that TTPL introduces to the POMO and LEHD on TSP1K. Subsequently, the projections learned from the POMO and LEHD were evaluated on the TSP1K to assess their large-scale generalization capabilities. The results are shown in Table A2.
>
> From these results, we can observe that while TTPL brings 4.29\% and 2.93\% additional computational overhead to POMO and LEHD, it yields significant performance improvements of 55.86\% and 58.89\% for each model on large-scale problems relative to their baseline performance, respectively.  **Consequently, we believe that this time investment is cost-effective**. Furthermore, we found that only using **half of the iteration budget can achieve the same results** as the original TTPL (compared to the results shown in Table 10, **A4**), which is a relatively small additional cost compared to the model training.
>
> Table A2: Computational overhead and performance enhancement introduced by TTPL on TSP1K instances
> ||Original||Original+TTPL||
> |-|-|-|-|-|
> |Method|TrainingTime|Performance|TrainingTime|Performance|
> |POMO|168h|43.51\%|175.2h(+7.2h)|24.35\%(-19.16\%)|
> |LEHD|188h|4.50\%|211h(+5.5h)|2.65\%(-1.85\%)|
>
> In short, our proposed TTPL method **requires only one-time learning** for a specific problem (e.g., TSP, CVRP) to generate the projection that is applicable across all other instances of that problem. Our experiment demonstrates that it is highly cost-effective. Thank you once again for your valuable feedback, and hope our response can address your concern.

---

### Official Review · Reviewer_gE2W · 2025-07-02

**Clarity:** 2
**Significance:** 2
**Originality:** 2
**Rating:** 4
**Confidence:** 2

**Summary:**

The paper proposes a framework to use LLMs to generate projection  strategies  that enhance model generalization for solving optimization problems.

**Questions:**

1. What are geometric transformations in Multi-View Decision Fusion?
2. Is the projection strategy a normalization operation?

**Ethical Concerns:**

["NO or VERY MINOR ethics concerns only"]

**Final Justification:**

Although there are still some issues, I am inclined to accept this paper.

**Limitations:**

yes

**Quality:**

2

**Strengths And Weaknesses:**

**Strengths:**

The proposed framework is interesting and seems to work well in practice.

**Weaknesses:**

1. The study lacks substantive novelty, with the core value of the methodology appearing to be primarily attributable to EoH [1].
2. The idea of mimicking evolutionary computation via LLMs is not new. In fact, most (all?) crossover and mutation operators are from EoH.
3. It remains unclear what specifically constitutes the NCO model in fitness evaluation and whether the selection thereof significantly impacts the results.
4. The geometric transformations in Multi-View Decision Fusion are not clearly described.
5. The adoption of LEHD [2] as the base model, while employing POMO and other NCO methods as baselines, appears somewhat inappropriate in the experimental design.

[1] Evolution of Heuristics: Towards Efficient Automatic Algorithm Design Using Large Language Model, ICML 2024.

[2] Neural combinatorial optimization with heavy decoder: Toward large scale generalization, NeurIPS 2023.

---

> ### Author Rebuttal · Authors · 2025-07-31
>
> Dear reviewer gE2W
>
> Thank you very much for your time and effort in reviewing our work. We are glad to know you find our proposed framework interesting and work well. We address your concerns as follows.
>
> > **W1. The study lacks substantive novelty, with the core value of the methodology appearing to be primarily attributable to EoH.**
>
> > **W2. The idea of mimicking evolutionary computation via LLMs is not new. In fact, most (all?) crossover and mutation operators are from EoH.**
>
> **A1:** Thanks very much for your comments. We would like to deal with your concerns from the following two aspects.
>
> **1. Novelty**
>
> We respectfully clarify that this work focuses on enhancing the large-scale generalization ability of NCO methods. Our novelty lies in the introduction of Test-Time Projection Learning (TTPL), a light-weight and universal strategy that can significantly enhance the NCO method's generalization performance.
> Unlike most existing works that focus on modifying model architectures or training paradigms [5-8], our TTPL framework takes a **data-centric** approach. It only operates at the inference stage by aligning the distributions of testing and training data.
>
> To demonstrate its universality across models, we have integrated TTPL into four different models: POMO [4], SIGD [7], BQ [6], and LEHD [5]. As shown in Table 3 (**A4**), the model equipped with TTPL remarkably outperforms the original version on most test instances.
> In addition, we would like to emphasize the generalization performance improvements that TTPL brings to NCO methods in large-scale scenarios. As reported in our main experiments (**Table 2, page 7**),  LEHD using the projection learned from TTPL achieves improvements of 14.79\%, 78.98\%, and 86.67\% on TSP instances of sizes 1K, 5K, and 10K, respectively. It is worth noting that it surpasses existing methods and obtains the SOTA results.
>
> **2. Independence on LLM-AHD methods**
>
> In this work, we do not intend to propose a new LLM-AHD method, but rather aim to employ LLM-AHD for our TTPL implementation. Our proposed TTPL framework is **independent of any specific LLM-AHD methods**. We empirically show that it is a replaceable component within our TTPL framework. To demonstrate it, we replace EoH with two alternative LLM-AHD methods, Funsearch [2] and ReEvo [3], to generate new projection strategies for the LEHD. As presented in Table 1, both resulting projections outperform the original LEHD, confirming the independence of TTPL on the LLM-AHD methods.
>
> Table 1: Results of TTPL using different LLM-AHD methods on TSP1K instances
> |Method|Obj.|
> |-|-|
> |w/o TTPL| 24.16|
> |TTPL-EoH|23.73|
> |TTPL-FunSearch|23.81|
> |TTPL-ReEvo|23.74|
>
> > **W3. It remains unclear what specifically constitutes the NCO model in fitness evaluation and whether the selection thereof significantly impacts the results.**
>
> **A2:** Thank you for your raising this important concern. We would like to answer your comments in two parts.
>
> **Explanation of fitness evaluation**
>
> In principle, the NCO model within the fitness evaluation phase can be any base model capable of solving the routing problem (e.g., TSP, CVRP). In the specific experiments presented in our study, we employed LEHD as the base model. The evaluation process is described below. First, TTPL proposes a candidate projection strategy. The strategy is applied to a test instance to align its data distribution with the training instances. The aligned test instance is then fed into the chosen base model (e.g., LEHD), which yields a solution. Finally, the quality of this solution is returned to TTPL as the fitness value for the corresponding projection strategy.
>
> **Sensitivity on different base models**
>
> To verify the impact of the selection of the base model, we adopt TTPL to two additional NCO models, namely POMO and BQ, as base models. The models together with developed TTPL strategies are then tested on the TSP1K instances, and the results are summarized in Table below. It demonstrates that the effectiveness of TTPL consistently exists across different base models
>
> Table 2: Results of TTPL using different base models on TSP1K instances
> ||TTPL-LEHD|TTPL-POMO|TTPL-BQ|
> |-|-|-|-|
> |base model w/o proj|24.16|33.18|23.65|
> |base model with proj|**23.73**|**28.75**|**23.57**|
>
> > **W4. The geometric transformations in Multi-View Decision Fusion are not clearly described.**
> > **Q1. What are geometric transformations in Multi-View Decision Fusion?**
>
> **A3:** We sincerely apologize for the lack of clarity in our original description of the geometric transformations and thank you for pointing this out, providing us with the opportunity to elaborate on the key component of our method. We will add a detailed description to our main text in the future.
>
> The geometric transformations used in our Multi-View Decision Fusion (MVDF) module are specifically a series of **rotational augmentation**, which is inspired by the augmentation mechanism introduced in POMO [4]. In particular, for a given test instance, we generate a projected K-Nearest-Neighbors (KNN) subgraph via the projection obtained in the TTPL. Then, the MVDF rotates the subgraph around its center by angles of $45^{\circ},90^{\circ},135^{\circ},180^{\circ},225^{\circ},270^{\circ},$ and $315^{\circ}$. Including the original subgraph, this process yields a total of **8 distinct geometric views**. By incorporating these views, MVDF is capable of enhancing the robustness of the final decision.
>
> > **W5. The adoption of LEHD [2] as the base model, while employing POMO and other NCO methods as baselines, appears somewhat inappropriate in the experimental design.**
>
> **A4:**  Thanks for your constructive comments. Following your insightful suggestion, we provide direct comparisons for NCO models, evaluating their performance both with and without the TTPL projection strategy. Specifically, we compare POMO [2], LEHD [5], BQ [6], and SIGD [7], four **constructive** models, with and without TTPL projection. The test instances are set to TSP1K/5K/10K with the same settings as in our original paper. The results are shown in the Table below
>
> Table 3: Results of different base models with and w/o TTPL
> |Method|TSP1K|TSP5K|TSP10K|
> |-|-|-|-|
> |POMO greedy|33.18(30.32\%)|94.04(45.79\%)|150.64(52.35\%)|
> |POMO-TTPL|**29.68(22.1\%)**|**71.76(28.97\%)**|**101.8(29.49\%)**|
> |SIGD greedy|**23.57(1.96\%)**|57.19(12.20\%)|93.80(30.68\%)|
> |SIGD-TTPL|23.86(3.1\%)|**56.05(9.06\%)**|**80.44(10.77\%)**|
> |BQ greedy|23.65(2.30\%)|58.27(14.31\%)|89.73(25.02\%)|
> |BQ-TTPL|**23.58(1.95\%)**|**56.47(9.74\%)**|**81.48(11.9\%)**|
> |LEHD greedy|23.84(3.11\%)|58.85(15.46\%)|91.33(27.24\%)|
> |LEHD-TTPL|**23.73(2.65\%)**|**52.63(3.25\%)**|**74.39(3.63\%)**|
>
> It clearly indicates that the TTPL consistently improves the performance of all base models. For every model and dataset except SIGD on TSP1K, the TTPL variant achieves a lower optimal gap compared to its original counterpart.
>
> > **Q2. Is the projection strategy a normalization operation?**
>
> **A5:** Thanks very much for your question, which allows us to clarify the relationship between the projection strategy and the normalization operation. **In our study, normalization is a subset of the projection strategy**. To elaborate on the distinction, **projection strategy**: Its goal is to align the distribution of an input instance with a target distribution. This mapping is highly flexible and can encompass a set of transformations, including scaling and rotation. **Normalization operation**: it involves **scaling** an instance's coordinates to fit within a unit interval (e.g., [0, 1]). This is a fixed operation that does not include transformations like rotation or translation. We are more than willing to present further clarification about the difference between the projection strategy and the normalization operation upon any specific remaining questions.
>
> Thank you for your time and effort dedicated to reviewing our work. We sincerely hope that our responses have effectively addressed your concerns.
>
> **Reference**
>
> [1] Liu, F., Tong, X., Yuan, M., Lin, X., Luo, F., Wang, Z., ... \& Zhang, Q. (2024). Evolution of Heuristics: Towards efficient automatic algorithm design using large language model. arXiv preprint arXiv:2401.02051.
>
> [2] Romera-Paredes, B., Barekatain, M., Novikov, A., Balog, M., Kumar, M. P., Dupont, E., ... \& Fawzi, A. (2024). Mathematical discoveries from program search with large language models. Nature, 625(7995), 468-475.
>
> [3] Ye, H., Wang, J., Cao, Z., Berto, F., Hua, C., Kim, H., ... \& Song, G. (2024). ReEvo: Large language models as hyper-heuristics with reflective evolution. Advances in neural information processing systems, 37, 43571-43608.
>
> [4] Kwon, Y. D., Choo, J., Kim, B., Yoon, I., Gwon, Y., \& Min, S. (2020). POMO: Policy optimization with multiple optima for reinforcement learning. Advances in Neural Information Processing Systems, 33, 21188-21198.
>
> [5] Luo, F., Lin, X., Liu, F., Zhang, Q., \& Wang, Z. (2023). Neural combinatorial optimization with heavy decoder: Toward large scale generalization. Advances in Neural Information Processing Systems, 36, 8845-8864.
>
> [6] Drakulic, D., Michel, S., Mai, F., Sors, A., \& Andreoli, J. M. (2023). BQ-NCO: Bisimulation quotienting for efficient neural combinatorial optimization. Advances in Neural Information Processing Systems, 36, 77416-77429.
>
> [7] Pirnay, J., \& Grimm, D. G. (2024). Self-improvement for neural combinatorial optimization: Sample without replacement, but improvement. arXiv preprint arXiv:2403.15180.
>
> [8] Fang, H., Song, Z., Weng, P., \& Ban, Y. (2024). INViT: A generalizable routing problem solver with invariant nested view transformer. arXiv preprint arXiv:2402.02317.

---

> > ### Comment · Reviewer_gE2W · 2025-08-02
> >
> > To authors,
> >
> > Thanks for your response. I feel that the work merely performs evolutionary updates on existing projection strategies without offering substantial new insights. I found that when the existing INViT is used as the projection strategy within the LEHD framework, your advantage is not very significant. Moreover, the best-performing method in the paper mainly relies on LEHD’s RRC, which is highly costly. How does your improved projection strategy compare when evaluated against LEHD RRC 1000 + INViT? Does it demonstrate a clear advantage?

---

> ### Author Response · Authors · 2025-08-05
> **Response to reviewer gE2W (1/2)**
>
> Dear Reviewer gE2W,
>
> Thank you for your continued engagement in the discussion. We sincerely appreciate your follow-up questions, which have given us a valuable opportunity to further clarify the contribution of our work. We address your concerns point-by-point as follows:
>
> > **1. Insight \& Advantage**
>
> We would like to clarify that the new insight provided by our work is our **test-time instance alignment approach**. It is designed to improve the large-scale generalization of NCO models via aligning test and training data distributions. To achieve this, we not only develop a learning framework through LLM to automatically learns the projection strategy, but also design a novel **(Multi-View Decision Fusion) MVDF** module to refine the initial alignment, providing robustness and ensuring final node selection. This hierarchical methodology allows the test instances to approximate the training distribution, thereby enhancing models' large-scale generalization abilities. The proposed methods have demonstrated their superior performance on TSP and CVRP and universality across multiple models (as shown in Table 2 in the main text, Table 3 in **A5**).
>
> > **1.1 Ablation Experiment Setting**
>
> Following your advice, we have conducted new experiments to systematically validate the effectiveness of the **TTPL projection**. In addition, we also evaluate the efficacy of our **MVDF refinement module**. The evaluation utilizes synthetic TSP and CVRP instances with 1K to 10K nodes and real-world TSPLIB/CVRPLIB datasets, under the same settings described in the main text. We report the results in Tables A1 and A2.
>
> > **1.2 Effectiveness of TTPL projection**
>
> From these results, we can observe that while our method yields slightly superior results on TSP, it demonstrates a clear advantage on CVRP. On CVRP1K, 5K, and 10K, LEHD+TTPL achieves 81.57\%, 79.14\%, and 78.07\% improvements in terms of optimality gap compared to LEHD+INViT when both methods use RRC1000. The performance on TSP suggests that the simplicity of TSP node distribution makes testing and training instances straightforward to align via projection. However, the geometric complexity increases in CVRP instances introduces a more significant distributional shift, causing the INViT's projection method to exhibit limited generalization during the testing phase.
>
> The limitation of INViT projection on CVRP highlights that its **one-time heuristic designs** prevent it from being effective across different problems. To overcome this, we leverage the LLM to automatically learn the problem-specific features and design corresponding strategies, enabling our approach to achieve consistently superior performance across different problems.
>
> > **1.3 Effectiveness of MVDF**
>
> Furthermore, from these results, we can also observe that in each tested scenario, LEHD with MVDF achieves the best-performing results. For example, on the TSPLIB and CVRPLIB benchmarks, LEHD+TTPL with MVDF improves performance by 33.1\% and 16.83\%, respectively, compared to LEHD+TTPL. This substantial improvement on TSP and CVRP underscores the necessity of applying a refinement step to the initial projection, especially in real-world scenarios.
>
> Table. A1: Results of LEHD using different projection strategies as well as the MVDF refinement module on synthetic TSP and CVRP datasets
> |Method|TSP1K|TSP5K|TSP10K|
> |-|-|-|-|
> |LEHD+INViT greedy|2.96\%|3.32\%|3.77\%|
> |LEHD+TTPL greedy|2.65\%|3.25\%|3.63\%|
> |LEHD+TTPL+MVDF greedy|**2.26\%**|**2.86\%**|**2.39\%**|
> ||||
> |**Method**|**TSP1K**|**TSP5K**|**TSP10K**|
> |LEHD+INViT RRC1000|0.33\%|1.32\%|1.47\%|
> |LEHD+TTPL RRC1000|0.33\%|1.29\%|1.42\%|
> |LEHD+TTPL+MVDF RC1000|**0.28\%**|**1.04\%**|**1.13\%**|
> ||||
> |**Method**|**CVRP1K**|**CVRP5K**|**CVRP10K**|
> |LEHD+INViT greedy|46.68\%|30.87\%|22.31\%|
> |LEHD+TTPL greedy|8.60\%|6.44\%|5.11\%|
> |LEHD+TTPL+MVDF greedy|**7.79\%**|**4.88\%**|**2.54\%**|
> ||||
> |**Method**|**CVRP1K**|**CVRP5K**|**CVRP10K**|
> |LEHD+INViT RRC1000|39.46\%|26.09\%|19.52\%|
> |LEHD+TTPL RRC1000|3.55\%|3.72\%|3.15\%|
> |LEHD+TTPL+MVDF RRC1000|**3.28\%**|**2.37\%**|**1.02\%**|
>
> Table. A2: Results of LEHD using different projection strategies as well as the MVDF refinement module on TSP/CVRPLIB instances.
> ||TSPLIB|||CVRPLIB|||
> |-|-|-|-|-|-|-|
> |Method|1K$<$n$\leq$5K|n$>$5K|All\#|1K$<$n$\leq$7K|n$>$7K|All\#|
> |LEHD+INViT|6.33\%|4.95\%|5.92\%|34.66\%|22.67\%|29.52\%|
> |LEHD+TTPL|6.04\%|4.57\%|5.80\%|11.89\%|14.93\%|13.19\%|
> |LEHD+TTPL+MVDF|**3.81\%**|**4.05\%**|**3.88\%**|**10.41\%**|**11.71\%**|**10.97\%**|

---

> ### Author Response · Authors · 2025-08-05
> **Response to reviewer gE2W (2/2)**
>
> > **2. Efficacy of Inference Strategy**
>
> We acknowledge that using RRC1000 is highly costly. However, as demonstrated in Table. A3, we find that our method using RRC70 can already outperform baseline methods on all TSP instances with acceptable time budgets. For example, on TSP5K, TTPL-MVDF with RRC70 takes 19.5 seconds to obtain a solution, while BQ bs16 takes 24 seconds to solve an instance. Moreover, we would like to clarify that our proposed TTPL-MVDF without RRC can also surpass the best-performed baseline models by 23.71\%, 7.94\%, 47.25\%, and 50.19\% on TSP5K, 10K, 50K, and 100K instances in terms of optimality gap.
>
> Table. A3: Comparison results on synthetic TSP dataset of TTPL-MVDF and TTPL-MVDF with RRC70
> |Method|TSP1K||TSP5K||TSP10K||TSP50K||TSP100K||
> |-|-|-|-|-|-|-|-|-|-|-|
> ||Obj.(Gap)|Time|Obj.(Gap)|Time|Obj.(Gap)|Time|Obj.(Gap)|Time|Obj.(Gap)|Time|
> |DIFUSCO*|23.39 (1.17\%)|11.5s|——|——|73.62 (2.58\%)|3.0m|——|——|——|——|
> |H-TSP|24.66 (6.66\%)|48s|55.16 (8.21\%)|1.2m|77.75 (8.38\%)|2.2m|OOM||OOM||
> |GLOP|23.78 (2.85\%)|10.2s|53.15(4.26\%)|1.0m|75.04(4.39\%)|1.9m|168.09(5.10\%)|1.5m|237.61(5.14\%)|3.9m|
> |POMO aug*8|32.51(40.6\%)|4.1s|87.72(72.1\%)|8.6m|OOM||OOM||OOM||
> |ELG aug*8|25.738(11.33\%)|0.8s|60.19(18.08\%)|21s|OOM||OOM||OOM||
> |LEHD RRC1,000|23.29(0.72\%)|3.3m|54.43(6.79\%)|8.6m|80.90(12.5\%)|18.6m|OOM||OOM||
> |BQ bs16|23.43(1.37\%)|13s|58.27(10.7\%)|24s|OOM||OOM||OOM||
> |SIGD bs16|23.36(1.03\%)|17.3s|55.77(9.42\%)|30.5m|OOM||OOM||OOM||
> |INViT-3V greedy|24.66(6.66\%)|9.0s|54.49(6.90\%)|1.2m|76.85(7.07\%)|3.7m|171.42(7.18\%)|1.3h|242.26(7.20\%)|5.0h|
> |LEHD greedy|23.84(3.11\%)|0.8s|58.85(15.46\%)|1.5m|91.33(27.24\%)|11.7m|OOM||OOM||
> |BQ greedy|23.65(2.30\%)|0.9s|58.27(14.31\%)|22.5s|89.73(25.02\%)|1.0m|OOM||OOM||
> |SIGD greedy|23.573(1.96\%)|1.2s|57.19(12.20\%)|1.8m|93.80(30.68\%)|15.5m|OOM||OOM||
> |TTPL-MVDF|23.64(2.26\%)|0.5s|52.43(3.25\%)|3.2s|73.49(2.39\%)|7.2s|164.24(2.69\%)|32.1s|231.78(2.56\%)|1.1m|
> |TTPL-MVDF RRC70|**23.28(0.69\%)**|2.3s|**51.78(1.58\%)**|19.5s|**72.96(1.64\%)**|24.5s|**163.74(2.38\%)**|50.6s|**231.4(2.39\%)**|1.4m|
>
> Thank you again for your valuable feedback. We hope that the response has successfully addressed your concerns.

---

> > ### Comment · Reviewer_gE2W · 2025-08-09
> >
> > Thank you for the additional response from the authors.
> >
> > **On "Insight & Advantage"**
> >
> >   * Is the current evolutionary validation step performed directly on the test set? Evolving a projection strategy at a significant cost may only result in a projection method tailored to the current test set, which does not necessarily indicate alignment with the training data distribution.
> >
> >   * (Multi-View Decision Fusion) MVDF has already been proposed in the POMO and is not considered a contribution of the current work.

---

> > ### Comment · Reviewer_gE2W · 2025-08-09
> >
> > Although there are still some issues, I am inclined to accept this paper.

---

> > > ### Author Response · Authors · 2025-08-09
> > > **Thank you very much**
> > >
> > > Thank you very much for your effort in reviewing our paper and engaging with us in the discussion. We are thrilled to know you are inclined to accept our paper.

---

> ### Author Response · Authors · 2025-08-09
> **Response to Reviewer gE2W (updated)**
>
> Dear Reviewer gE2W
>
> Thank you for your continued questions. We address your concerns point-by-point as follows:
>
> > **1. Projection strategy design**
>
> We would like to clarify that the TTPL designs a projection strategy using **an independent validation dataset** that is kept strictly separate from the data used for final evaluation. To assess its generalization ability, we then evaluated this strategy on a diverse range of unseen instances, including synthetic TSP/CVRP problems with various node distributions and real-world benchmarks from TSP/CVRPLib (you may kindly refer to **Table 11 in Appendix D.2 and Table 3 in the main text**). The method's strong performance across these instances confirms its effectiveness in enhancing NCO methods' large-scale generalization capabilities.
>
> > **2. MVDF vs. POMO**
>
> We would like to clarify that MVDF and POMO are two different strategies, with the specific distinctions being as follows:
>
> **Mechanism:** POMO's augmentation is only applied at the beginning of the solving process. Our MVDF, conversely, operates at each step of the node selection process.  It performs after our learned projection maps a subgraph back to a distribution similar to the training instances. In such cases, MVDF can effectively address potential local density variations within the transformed subgraph, thereby facilitating robust decisions. As demonstrated in Table B1, our MVDF method outperforms POMO's augmentation strategy on large-scale instances, which substantiates the superior advantage of our policy enhancement mechanism in tackling large-scale challenges.
>
> **Modularity**: We have merely leveraged POMO's data transformation strategy to generate "views," and this component is replaceable. In principle, any transformation method capable of providing different observational angles of a subgraph can be integrated into our MVDF framework. To validate this, we use the augmentation strategy from INViT to replace the original POMO augmentation, and show the results in Table B2. From these reulsts, we can observe that MVDF with INViT augmentation outperforms its original version, confirming that POMO augmentation is not necessary in the MVDF module.
>
> Table. B1: Results of LEHD using POMO augmentation and MVDF on large-scale TSP and CVRP instances
> |Method|CVRP10K|CVRP50K|CVRP100K|
> |-|-|-|-|
> |LEHD-pomo aug|3.52\%|4.27\%|4.77\%|
> |LEHD-MVDF|**2.54\%**|**2.73\%**|**3.60\%**|
>
> Table. B2: Results of MVDF using different data augmentation techniques on CVRP instances
> | Method                    | CVRP50K | CVRP100K |
> |---------------------------|---------|----------|
> | LEHD-MVDF with POMO aug 8 | 2.73\%  | 3.60\%   |
> | LEHD-MVDF with INViT aug   | **1.90\%**  | **1.85\%**   |

---

### Decision · Program_Chairs · 2025-09-17

**Decision:**

Accept (poster)

**Comment:**

This paper focuses on LLM-based test-time methods for combinatorial optimization. All reviewers unanimously agree that this paper makes significant contributions and recommend acceptance (4,4,4,4). After a careful evaluation, the AC concurs with the reviewers’ assessments and also recommends acceptance.